# A systematic review of worldwide causal and correlational evidence on digital media and democracy

Philipp Lorenz-Spreen [1,5] ✉, Lisa Oswald [2,5], Stephan Lewandowsky [3,4] & Ralph Hertwig [1]

One of today's most controversial and consequential issues is whether the global uptake of digital media is causally related to a decline in democracy. We conducted a systematic review of causal and correlational evidence ($N = 496$ articles) on the link between digital media use and different political variables. Some associations, such as increasing political participation and information consumption, are likely to be beneficial for democracy and were often observed in autocracies and emerging democracies. Other associations, such as declining political trust, increasing populism and growing polarization, are likely to be detrimental to democracy and were more pronounced in established democracies. While the impact of digital media on political systems depends on the specific variable and system in question, several variables show clear directions of associations. The evidence calls for research efforts and vigilance by governments and civil societies to better understand, design and regulate the interplay of digital media and democracy.

The ongoing heated debate on the opportunities and dangers that digital media pose to democracy has been hampered by disjointed and conflicting results (for recent overviews, see refs. [1–4]). Disagreement about the role of new media is not a novel phenomenon; throughout history, evolving communication technologies have provoked concerns and debates. One likely source of concern is the dual-use dilemma, that is, the inescapable fact that technologies can be used for both noble and malicious aims. For instance, during the Second World War, radio was used as a propaganda tool by Nazi Germany[5], whereas allied radio, such as the BBC, supported resistance against the Nazi regime, for example, by providing tactical information on allied military activities[6,7]. In the context of the Rwandan genocide, radio was used to incite Rwandan Hutus to massacre the country's Tutsi minority[8]. In the aftermath of the genocide, using the same means to cause different ends, the radio soap opera 'Musekeweya' successfully reduced intergroup prejudice in a year-long field experiment[9,10].

Digital media appears to be another double-edged sword. On the one hand, it can empower citizens, as demonstrated in movements such as the Arab Spring[11], Fridays for Future and #MeToo[12]. On the other hand, digital media can also be instrumental in inciting destructive behaviours and tendencies such as polarization and populism[13], as well as fatal events such as the attack on the United States Capitol in January 2021. Relatedly, the way political leaders use or avoid digital media can vary greatly depending on the political context. Former US President Trump used it to spread numerous lies ranging from claims about systematic voter fraud in the 2020 presidential election to claims about the harmlessness of Covid-19. In spring 2022, Russian President Putin had banned most social media platforms that would bypass the state-controlled classical media, probably to prevent access to information about his army's attack on Ukraine[14]. At the same time, Ukrainian President Zelensky has skilfully used social media to boost Ukrainian morale and engage in the information war with Russia. Examples of the dual-use dilemma of digital media abound.

[1]Center for Adaptive Rationality, Max Planck Institute for Human Development, Berlin, Germany. [2]Hertie School, Berlin, Germany. [3]School of Psychological Science and Cabot Institute, University of Bristol, Bristol, UK. [4]School of Psychological Science, University of Western Australia, Perth, Australia. [5]These authors contributed equally: Philipp Lorenz-Spreen and Lisa Oswald. ✉e-mail: lorenz-spreen@mpib-berlin.mpg.de

Clearly, digital media can foster liberation, democratization and participation, but can also play an important role in eroding democracy. The role of digital media is further complicated because unlike other communication technologies, it enables individuals to easily produce and disseminate content themselves, and offers largely frictionless interaction between users. These properties have not only moved the self-organized political behaviour of citizens into the spotlight[15], but have also shifted power to large digital media platforms. Unlike broadcasters, digital media platforms typically do not create content; instead, their power lies in providing and governing a digital infrastructure. Although that infrastructure could serve as an online public sphere[16], it is the platforms that exert much control over the dynamics of information flow.

Our goal is to advance the scientific and public debate on the relationship between digital media and democracy by providing an evidence-based picture of this complex constellation. To this end, we comprehensively reviewed and synthesized the available scientific knowledge[17] on the link between digital media and various politically important variables such as participation, trust and polarization.

We aimed to answer the pre-registered question "If, to what degree and in which contexts, do digital media have detrimental effects on democracy?" (pre-registered protocol, including research question and search strategy, at https://osf.io/7ry4a/). This two-stage question encompasses, first, the assessment of the direction of effects and, second, how these effects play out as a function of political contexts.

A major difficulty facing researchers and policymakers is that most studies relating digital media use to political attitudes and behaviours are correlational. Because it is nearly impossible to simulate democracy in the laboratory, researchers are forced to rely on observational data that typically only provide correlational evidence. We therefore pursued two approaches. First, we collected and synthesized a broad set of articles that examine associations between digital media use and different political variables. We then conducted an in-depth analysis of the small subset of articles reporting causal evidence. This two-step approach permitted us to focus on causal effects while still taking the full spectrum of correlational evidence into account.

For the present purpose, we adopted a broad understanding of digital media, ranging from general internet access to the use of specific social media platforms, including exposure to certain types of content on these platforms. To be considered as a valid digital media variable in our review, information or discussion forums must be hosted via the internet or need to describe specific features of online communication. For example, we considered the online outlets of traditional newspapers or TV channels as digital source of political information but not the original traditional media themselves. We provide an overview of digital media variables present in our review sample in Fig. 1d and discriminate in our analyses between the two overarching types of digital media: internet, broadly defined, on the one hand and social media in particular on the other hand.

We further aimed to synthesize evidence on a broad spectrum of political attitudes and behaviours that are relevant to basic democratic principles[18]. We therefore grounded our assessment of political variables in the literature that examines elements of modern democracies that are considered essential to their functioning, such as citizens' basic trust in media and institutions[19], a well-informed public[20], an active civil society[21,22] and exposure to a variety of opinions[23,24]. We also included phenomena that are considered detrimental to the functioning of democracies, including open discrimination against people[25], political polarization to the advantage of political extremists and populists[26] and social segregation in homogeneous networks[23,27].

The political variables in focus are themselves multidimensional and may be heterogeneous and conflicting. For example, polarization encompasses partisan sorting[28], affective polarization[29], issue alignment[30,31] and a number of other phenomena (see ref. [32] for an excellent literature review on media effects on variations of ideological and affective polarization). For our purpose, however, we take a broader perspective, examining and comparing across different political variables the directions—beneficial or detrimental to democracy—in which digital media effects play out.

Notwithstanding the nuances within each dimension of political behaviour, wherever possible we explicitly interpreted each change in a political variable as tending to be either beneficial or detrimental to democracy. Even though we tried to refrain from normative judgements, the nature of our research question required us to interpret the reported evidence regarding its relation to democracy. For example, an increase in political knowledge is generally considered to be beneficial under the democratic ideal of an informed citizenry[20]. Similarly, a certain level of trust in democratic institutions is crucial for a functioning democracy[33]. By contrast, various forms of polarization (particularly affective polarization) tend to split societies into opposing camps and threaten democratic decision-making[34,35]. Likewise, populist politics that are often coupled with right-wing nationalist ideologies, artificially divide society into a corrupt 'elite' that is opposed by 'the people', which runs counter to the ideals of a pluralistic democracy and undermines citizens' trust in politics and the media[36,37]. We therefore considered polarization and populism, for example, to be detrimental to democracy.

There are already some systematic reviews of subsets of associations between political behaviour and media use that fall within the scope of our analysis, including reviews of the association between media and radicalization[38,39], polarization[32], hate speech[40], participation[41–45], echo chambers[46] and campaigning on Twitter[47]. These extant reviews, however, did not contrast and integrate the wide range of politically relevant variables into one comprehensive analysis—an objective that we pursue here. For the most relevant review articles, we matched the references provided in them with our reference list (see Materials and Methods for details). Importantly, and unlike some extant reviews, our focus is not on institutions, the political behaviour of political elites (for example, their strategic use of social media; see refs. [47,48]), or higher-level outcomes (for example, policy innovation in governments[49]). We also did not consider the effects of traditional media (for example, television or radio) or consumption behaviours that are not specific to digital media (for example, selective exposure[50]). Furthermore, we did not focus on the microscopic psychological mechanisms that could shape polarization on social media (for a review, see ref. [51]). For reasons of external validity, we omitted small-scale laboratory-only experiments (for example, see ref. [52]), but included field experiments in our review. We included studies using a variety of methods—from surveys to large-scale analyses of social media data—and across different disciplines that are relevant to our research question. Details on the inclusion and exclusion criteria are provided in Materials and Methods. Our goal for this knowledge synthesis is to provide a nuanced foundation of shared facts for a constructive stage in the academic but also societal debate about the future of digital media and their role in democracy. In our view, this debate and the future design of digital media for democracy require a comprehensive assessment of its impact. We therefore not only focus on individual dimensions of political behaviour but also compare these dimensions and the methods by which they have been researched so far, thus going beyond the extant reviews. This approach aims to stimulate research that fills evidence gaps and establishes missing links that only become apparent when comparing the dimensions.

## Results

After conducting a pre-registered search (most recent update 15 September 2021) and selection process, we arrived at a final sample of $N = 496$ articles. For further analysis, we classified them by the set of variables between which they report associations: type of digital media (for example, social media, online news), political variables (for example, trust, participation) and characteristics of the information ecology

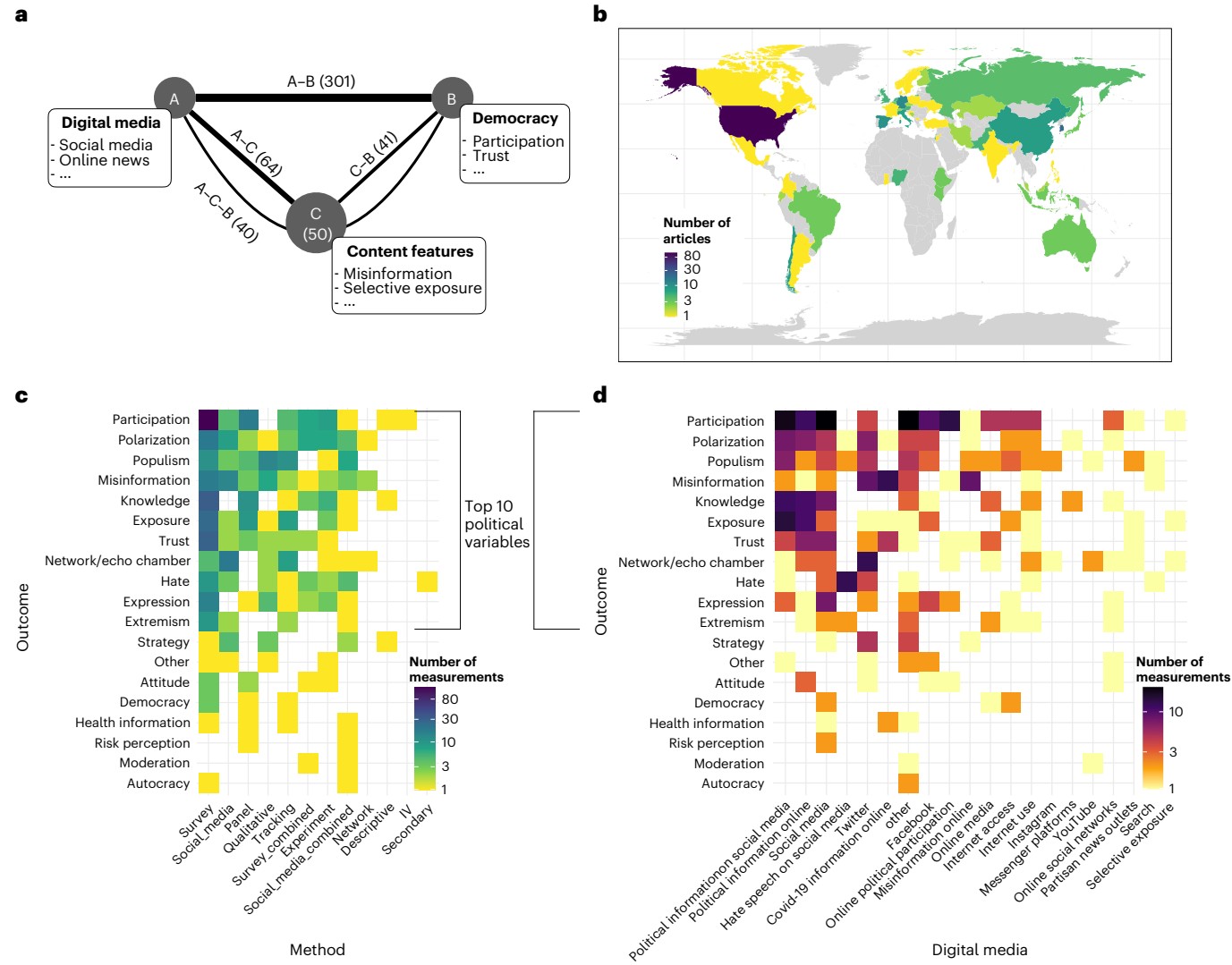

**Fig. 1 | Summary of the reviewed articles. a**, Combinations of variables in the sample: digital media (A), political variables (B) and content features such as selective exposure or misinformation (C). Numbers in brackets count articles in our sample that measure an association between variables. **b**, Geographic distribution of articles that reported site of data collection. **c**,**d**, Distribution of measurements (counted separately whenever one article reported several variables) over combinations of outcome variables and methods (**c**) and over combinations of outcome variables and digital media variables (**d**).

(for example, misinformation, selective exposure), as depicted in Fig. 1a. Each article was coded according to the combination of these variables as well as the method, specific outcome variable and, if applicable, the direction of association and potential moderator variables (see Materials and Methods for details). The resulting table of the fully coded set of studies can be found at https://osf.io/7ry4a/, alongside the code for the analyses and visualizations offered here.

Figure 1 reports the composition of the set of included articles. Figure 1a confirms that the search query mainly returned articles concerned with the most relevant associations between digital media and political outcomes. Most of the articles were published in the last 5 years, highlighting the fast growth of interest in the link between digital media and democracy. Articles span a range of disciplines, including political science, psychology, computational science and communication science. Although a preponderance of articles focused on the United States, there was still a large geographical variation overall (see Fig. 1b).

Figure 1c shows the distribution of measurements (counted separately when one article reported several outcomes) across methods and political variables. Our search query was designed to capture a broad

range of politically relevant variables, which meant that we had to group them into broader categories. The ten most frequently reported categories of variables were trust in institutions, different variants of political participation (for example, voter turnout or protest participation), exposure to diverse viewpoints in the news, political knowledge, political expression, measures of populism (for example, support for far-right parties or anti-minority rhetoric), prevalence and spread of misinformation, measures of polarization (for example, negative attitudes towards political opponents or fragmented and adversarial discourse), homophily in social networks (that is, social connections between like-minded individuals) and online hate (that is, hate speech or hate crime). Similarly, the distribution of outcomes and associated digital media variables in Fig. 1d shows that many studies focused on political information online, and specifically political information on social media, in combination with political polarization and participation, while other digital media variables, such as messenger platforms are less explored. The full table, including the reported political variables within each category, can be found at https://osf.io/7ry4a/. Figure 1 also reveals gaps in the literature, such as rarely explored geographical regions (for example, Africa) and under-studied methods–variable

combinations (for example, involving the combination of data sources such as social media data with survey or secondary data).

## Direction of associations

In the first part of our research question, we ask whether the available evidence suggests that the effects of digital media are predominantly beneficial or detrimental to democracy. To find an answer, we first selected subsets of articles that addressed the ten most frequently studied categories of political variables (hereafter simply referred to as political variables). We did not test specific hypotheses in our review. A total of $N$ = 354 associations were reported for these variables (when an article examined two relevant outcome variables, two associations were counted). The independent variable across these articles was always a measure of the usage of some type of digital media, such as online news consumption or social media uptake. Statistically speaking, the independent variables can be positively or negatively associated with the political outcome variable. For instance, more digital media use could be associated with more expression of hate (positive association), less expression of hate (negative association), or not associated at all. We decided to present relationships not at a statistical level but at a conceptual level. We therefore classified each observed statistical association as beneficial or detrimental depending on whether its direction was aligned or misaligned with democracy. For example, a positive statistical association between digital media use and hate speech was coded as a detrimental association; by contrast, a positive statistical association between digital media use and participation was coded as beneficial. Throughout, we represent beneficial associations in turquoise and detrimental associations in orange, irrespective of the underlying statistical polarity.

Figure 2 provides an overview of the ten most frequently studied political variables and the reported directions—colour-coded in terms of whether they are beneficial or detrimental to democracy—of each of their associations with digital media use. This overview encompasses both correlational and causal evidence. Some findings in Fig. 2 suggest that digital media can foster democratic objectives. First, the associations reported for participation point mostly in beneficial directions for democracy (aligned with previous results[45]), including a wide range of political and civic behaviour (Fig. 1d), from low-effort participation such as liking/sharing political messages on social media to high-cost activities such as protesting in oppressive regimes. Second, measures of political knowledge and diversity of news exposure appear to be associated with digital media in beneficial ways, but the overall picture was slightly less clear. Third, the literature is also split on how political expression is associated with digital media. Articles reporting beneficial associations between digital media and citizens' political expression were opposed by a number of articles describing detrimental associations. These detrimental associations relate to the 'spiral of silence' idea, that is, the notion that people's willingness to express their political opinions online depends on the perceived popularity of their opinions (see relevant overview articles[53,54]).

Fourth, we observed consistent detrimental associations for a number of variables. Specifically, the associations with trust in institutions were overwhelmingly pointing in directions detrimental to a functioning democracy. Measures of hate, polarization and populism were also widely reported to have detrimental associations with digital media use in the clear majority of articles. Likewise, increased digital media use was often associated with a greater exposure to misinformation. Finally, we also found that digital media were associated with homophily in social networks in detrimental ways (mostly measured on social media, and here especially on Twitter), but the pattern of evidence was a little less consistent. Differences in the consistency of results were also reflected when broken down along associated digital media variables (see insets in Fig. 2). For instance, both trust and polarization measures were consistently associated with media use across types of digital media ranging from social media to political

information online; in contrast, results for homophily were concentrated on social media and especially on Twitter, while measurements of news exposure were mostly concentrated on political information online. This points not only to different operationalizations of related outcome measures, such as diverse information exposure and homophilic network structures, but also to differences between the distinct domains of digital media in which these very related phenomena are measured. Similar observations can be made when separating associations between general types of digital media: social media vs internet more broadly (Supplementary Fig. 1).

Next, we distinguished between articles reporting correlational versus causal evidence and focused on the small subset of articles reporting the latter ($N$ = 24). We excluded causal evidence on the effects of voting advice applications from our summary as a very specific form of digital media, explicitly constructed to inform vote choices, and already extensively discussed in a meta-analysis[55].

## Causal inference

Usually, the absence of randomized treatment assignment, an inescapable feature of observational data (for example, survey data), precludes the identification of causal effects because individuals differ systematically on variables other than the treatment (or independent) variable. However, under certain conditions, it is possible to rule out non-causal explanations for associations, even in studies without random assignment that are based on observational data (see refs. [56–58]). For a more detailed explanation of the fundamental principles of causal inference, see Supplementary Material page 5 and, for example, the work of the 2021 laureates of the Nobel Memorial Prize in Economics[56–58].

Common causal inference techniques that were used in our sample include instrumental variable designs that introduce exogenous variation in the treatment variable[59–63], matching approaches to explicitly balance treatment and control groups[64–66], and panel designs that account for unobserved confounders with unit and/or time-fixed effects[67,68]. We also found multiple large-scale field experiments conducted on social media platforms[69–72] as well as various natural experiments[59,61,62,73].

Figure 3 summarizes the findings and primary causal inference techniques of these articles. Again, causal effects were coded as beneficial for or detrimental to democracy. This figure is structured according to whether evidence stemmed from established democracies or from emerging democracies and authoritarian regimes, adopting classifications from the Liberal Democracy Index provided by the Varieties of Democracy project[18]. In some autocratic regimes (for example, China), it is particularly difficult to interpret certain effects. For example, a loss of trust in government suggests a precarious development for an established democracy; in authoritarian regimes, however, it may indicate a necessary step toward overcoming an oppressive regime and, eventually, progressing towards a more liberal and democratic system. Instead of simply adopting the authors' interpretation of the effects or imposing our own interpretation of effects in authoritarian contexts, we leave this interpretation to the reader (denoted in purple in the figure). The overall picture converges closely with the one drawn in Fig. 2. We found general trends of digital media use increasing participation and knowledge but also increasing political polarization and decreasing trust that mostly aligned with correlational evidence.

## Effects on key political variables

In the following sections, we provide a short synopsis of the results, point to conflicting trends and highlight some examples of the full set of correlational and causal evidence, reported in Figs. 2 and 3, for six variables that we found to be particularly crucial for democracy: participation, trust, political knowledge, polarization, populism, network structures and news exposure. The chosen examples are stand-ins and illustrations of the general trends.

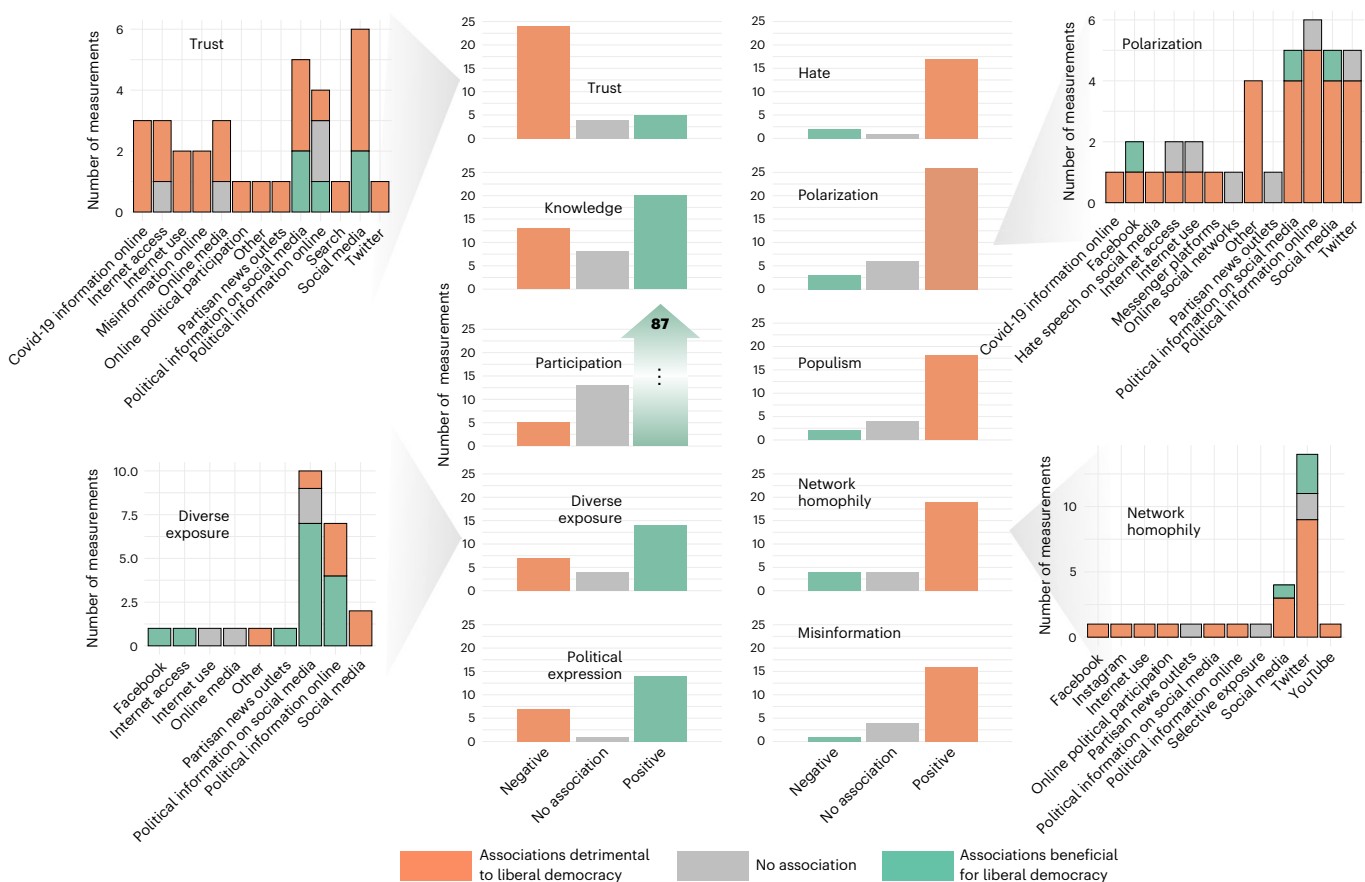

**Fig. 2 | Distribution of directions of associations from the full sample.** Directions of associations are reported for various political variables (see Fig. 1d for a breakdown). Insets show examples of the distribution of associations with trust, news exposure, polarization and network homophily over the different digital media variables with which they were associated.

**Participation.** Consistent with past meta-analyses[42,43,45], the body of correlational evidence supported a beneficial association between digital media use and political participation and mobilization.

Causal analyses of the effects of digital media on political participation in established democracies mostly studied voting and voter turnout[64,67,71,74–76]; articles concerned with other regions of the world rather focused on political protest behaviour[59,61,66]. Other articles considered online political participation[65,71]. One study, applying causal mediation analysis to assess a causal mechanism[77], found that information-oriented social media use affects political participation, mediated or enabled through the user's online political efficacy[65]. Overall, our evidence synthesis found largely beneficial mobilizing effects for political participation across this set of articles. Our search did not identify any studies that examined causal effects of digital media on political participation in authoritarian regimes in Africa or the Middle East.

**Trust.** Many articles in our sample found detrimental associations between digital media and various dimensions of trust (Fig. 2). For example, detrimental associations were found for trust in governments and politics[59,60,66,78–82], trust in media[83], and social and institutional trust[84]. During the COVID-19 pandemic, digital media use was reported to be negatively associated with trust in vaccines[85,86]. Yet the results about associations with trust are not entirely homogeneous. One multinational survey found beneficial associations with trust in science[87]; others found increasing trust in democracy with digital media use in Eastern and Central European samples[88,89]. Nevertheless, the large majority of reported associations between digital media use and trust

appear to be detrimental for democracy. While the evidence stems mostly from surveys, results gathered with other methods underpin these findings (Fig. 2 inset).

The majority of articles identifying causal effects also find predominantly detrimental effects of digital media on trust. A field experiment in the United States that set browser defaults to partisan media outlets[37] found a long-term loss of trust in mainstream media. Studies examining social trust as a central component of social capital find consistent detrimental effects of social media use[84]; in contrast, no effects of broadband internet in general on social trust was found[90]. In authoritarian regimes in Asia, increasing unrestricted internet access decreased levels of trust in the political system[59,73,91]. This finding confirms the predominant association observed in most other countries. Yet it also illustrates how digital media is a double-edged sword, depending on the political context: by reducing trust in institutions, digital media can threaten existing democracies as well as foster emerging democratic developments in authoritarian regimes.

**Political knowledge.** The picture was less clear for associations between the consumption of digital media and political knowledge. Still, the majority of associations point in beneficial directions and were found in both cross-sectional surveys[92–99] and panel surveys[100–102]. Studies linking web-tracking and survey data showed increased learning about politics[103], but also a turning away from important topics[104], whereas other experiments demonstrated an overall beneficial effect of digital media on issue salience[105]. These findings, however, stand in contrast to other studies that find a detrimental association between political knowledge and digital media use[106–110].

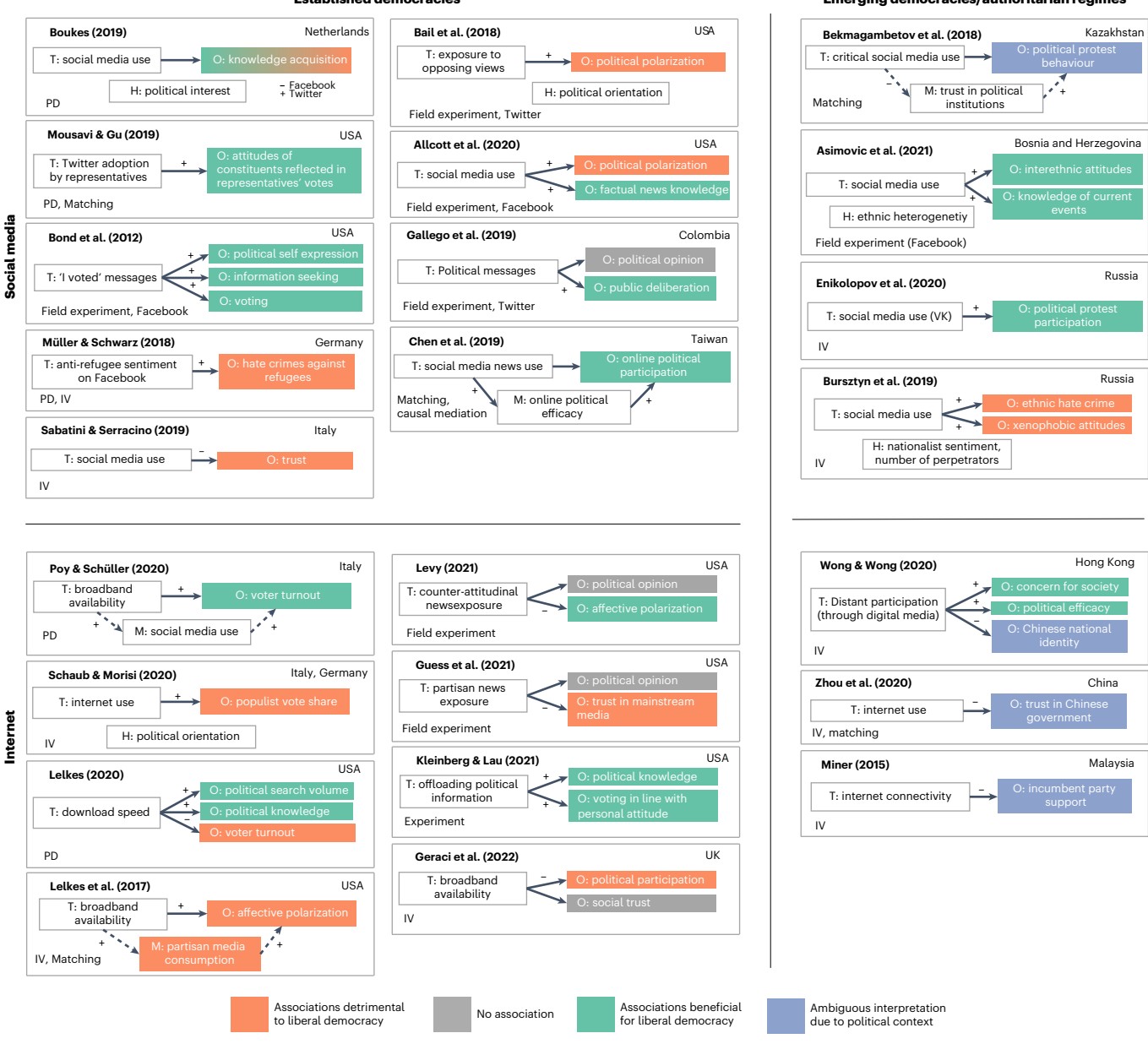

**Fig. 3 | Summary of causal evidence for digital media effects on political variables.** Each box represents one article. Treatments (T) are in white boxes on the left, political outcome (O) variables in coloured boxes on the right; M denotes mediators; H represents sources of effect heterogeneity or moderators. Positive (+) and negative (−) signs at paths indicate reported direction of effects. Location of sample indicated in top right corner of boxes, primary causal inference strategy in bottom left. Strategies include statistical estimation strategies such as instrumental variables (IV), matching and panel designs (PD) that use, for example, fixed effects (FE) or difference in difference (DiD) for causal estimation, as well as lab or field experiments (for example, field experiments rolled out on various platforms that are often supplemented with IV estimation to account for imperfect compliance). Detrimental effects on liberal democracy are shown in orange, beneficial effects in turquoise, effects open to interpretation in purple and null effects in grey. Solid arrows represent pathways for which authors provide causal identification strategies, dashed arrows represent descriptive (mediation) pathways.

The body of causal evidence on political knowledge also tends to paint a relatively promising picture. Multiple articles found that engagement with digital media increased political knowledge[67,70,72,74] and that engagement with political content on social media increased political interest among adolescents[111]. In line with these findings, it has been reported that political messages on social media, as well as faster download speed, can increase information-seeking in the political domain[67,71]. By contrast, there is evidence for a decrease in political knowledge[112], which is mediated through the news-finds-me effect: social media users believe that actively seeking out news is no longer required to stay informed, as they expect to be presented with important information.

It is important to note that most of these effects are accompanied by considerable heterogeneity in the population that benefits and the type of digital media. For example, politically interested individuals showed higher knowledge acquisition when engaging with Twitter, whereas the opposite effects emerged for engagement with Facebook[113]. Furthermore, there is evidence that the news-finds-me effect on social media can be mitigated when users consult alternative news sources[112].

**Polarization.** Most articles found detrimental associations between digital media and different forms of political polarization[114–118]. Our review obtained evidence for increasing outgroup polarization on social media in a range of political contexts and on various platforms[119–122]. Increasing polarization was also linked to exposure to viewpoints opposed to one's own on social media feeds[69,123]. Articles comparing several political systems found associations that were country-dependent[124], again highlighting the importance of political context[125]. Nevertheless, high digital media use was for the most part linked to higher levels of polarization, although there was some evidence for balanced online discourse without pronounced patterns of polarization[126–128], as well as evidence for potentially depolarizing tendencies[129].

The body of causal articles largely supported the detrimental associations of digital media that emerged, by and large, in the correlational articles. Among established democracies, both social media use and overall internet use increased political polarization[63,70]. This was also the case for an experimental treatment that exposed users to opposing views on Twitter[69]. However, some findings run counter to the latter result[130]: in a 2 month field experiment, exposure to counter-attitudinal news on Facebook reduced affective polarization (the authors used opposing news outlets as treatment instead of opinions on social media). Furthermore, one other field experiment did not find evidence that exposure to partisan online news substantively shifted political opinions but found a long-term loss of trust in mainstream media[37]. Still, taking all evidence into account, the overall picture remains largely consistent on the detrimental association between digital media and political polarization, including some but not all causal evidence.

**Populism.** Articles on populism in our review examined either vote share and other popularity indicators for populist parties or the prevalence of populist messages and communication styles on digital media. Overall, articles using panel surveys, tracking data and methods linking surveys to social media data consistently found that digital media use was associated with higher levels of populism. For example, digital platforms were observed to benefit populist parties more than they benefit established politicians[131–134]. In a panel survey in Germany, a decline in trust that accompanied increasing digital media consumption was also linked to a turn towards the hard-right populist AfD party[80]. This relationship might be connected to AfD's greater online presence, relative to other German political parties[132], even though these activities might be partly driven by automated accounts. There is also evidence for an association between increased social media use and online right-wing radicalization in Austria, Sweden and Australia[135–137]. Only a minority of articles found no relationship or the reverse relationship between digital media and populism[138–140]. For instance, in Japan, internet exposure was associated with increased tolerance towards foreigners[141].

Similarly, most causal inference studies linked increased populism to digital media use. For instance, digital media use in Europe led to increased far-right populist support[63,142], and there was causal evidence that digital media can propagate ethnic hate crimes in both democratic and authoritarian countries[62,68]. Leaving the US and European political context, in Malaysia, internet exposure was found to cause decreasing support for the authoritarian, populist government[60].

**Echo chambers and news exposure.** The evidence on echo chambers points in different directions depending on the outcome measure. On the one hand, when looking at news consumption, several articles showed that social media and search engines diversify people's news diets[67,143–146]. On the other hand, when considering social networks and the impact of digital media on homophilic structures, the literature contains consistent reports of ideologically homogeneous social clusters[147–151]. This underscores an important point: some seemingly paradoxical results can potentially be resolved by looking more closely at context and specific outcome measurement (see also Supplementary

Fig. 2). The former observation of diverse news exposure might fit with the beneficial relationship between digital media and knowledge reported in refs. [67,74,94,95,102], and the homophilic social structures could be connected to the prevalence of hate speech and anti-outgroup sentiments[120,152–155].

## Heterogeneity

We now turn to the second part of our research question and analyse the effects of digital media use in light of different political contexts. Figure 4 shows the geographical distribution of effect directions around the globe. Notably, most beneficial effects on democracy were found in emerging democracies in South America, Africa and South Asia. Mixed effects, by contrast, were distributed across Europe, the United States, Russia and China. Similarly, detrimental outcomes were mainly found in Europe, the United States and partly Russia, although this may reflect a lack of studies undertaken in authoritarian contexts. These patterns are also shown in Fig. 4c,d, where countries are listed according to the Liberal Democracy Index. Moderators—variables such as partisanship and news consumption that are sources of effect heterogeneity—displayed in Supplementary Fig. 3 also show slight differences between outcomes. Beneficial outcomes seemed to be more often moderated by political interest and news consumption, whereas detrimental outcomes tended to be moderated by political position and partisanship.

Furthermore, many causal articles acknowledge that effects differ between subgroups of their sample when including interaction terms in their statistical models. For example, the polarizing effects of digital media differ between Northern and Southern European media systems[142]: while consumption of right-leaning digital media increased far-right votes, especially in Southern Europe, the consumption of news media and public broadcasting in Northern European media systems with high journalistic standards appears to mitigate these effects. Another example of differential effects between subgroups was found in Russia, where the effects of social media on xenophobic violence were only present in areas with pre-existing nationalist sentiment. This effect was especially pronounced for hate crimes with a larger number of perpetrators, indicating that digital media was serving a coordinating function. In summary, a range of articles found heterogeneity in effects for varying levels of political interest[67,113], political orientation[63,69,70] and different characteristics of online content[111].

Most authors, particularly those of the causal inference articles in our body of evidence, explicitly emphasized the national, cultural, temporal and political boundary conditions for interpreting and generalizing their results (see, for example, ref. [111]). By contrast, especially in articles conducted on US samples, the national context and the results' potential conditionality was often not highlighted. We strongly caution against a generalization of findings that are necessarily bound to a specific political setting (for example, the United States) to other contexts.

## Sampling methods and risk of bias

To assess study quality and risk of bias, we additionally coded important methodological aspects of the studies, specifically, the sampling method, sample size and transparency indicators, such as competing interest, open data practices and pre-registrations. In Fig. 5, we show an excerpt from that analysis. Different sampling methods naturally result in different sample sizes as shown in Fig. 5a,b. Furthermore, behavioural data are much more prevalent for studies that look at detrimental outcomes, such as polarization and echo chambers. Classic surveys with probability samples or quota samples, in contrast, are often used to examine beneficial outcome measures such as trust and participation (Fig. 5c,d). Overall, however, no coherent pattern emerges in terms of the reported directions of associations. If anything, large probabilistic samples report relatively less beneficial associations for both types of outcomes (Fig. 5). Generally, different types of data have different advantages, such as probability and quota samples approximating more closely the ideal of representativeness, whereas the observation

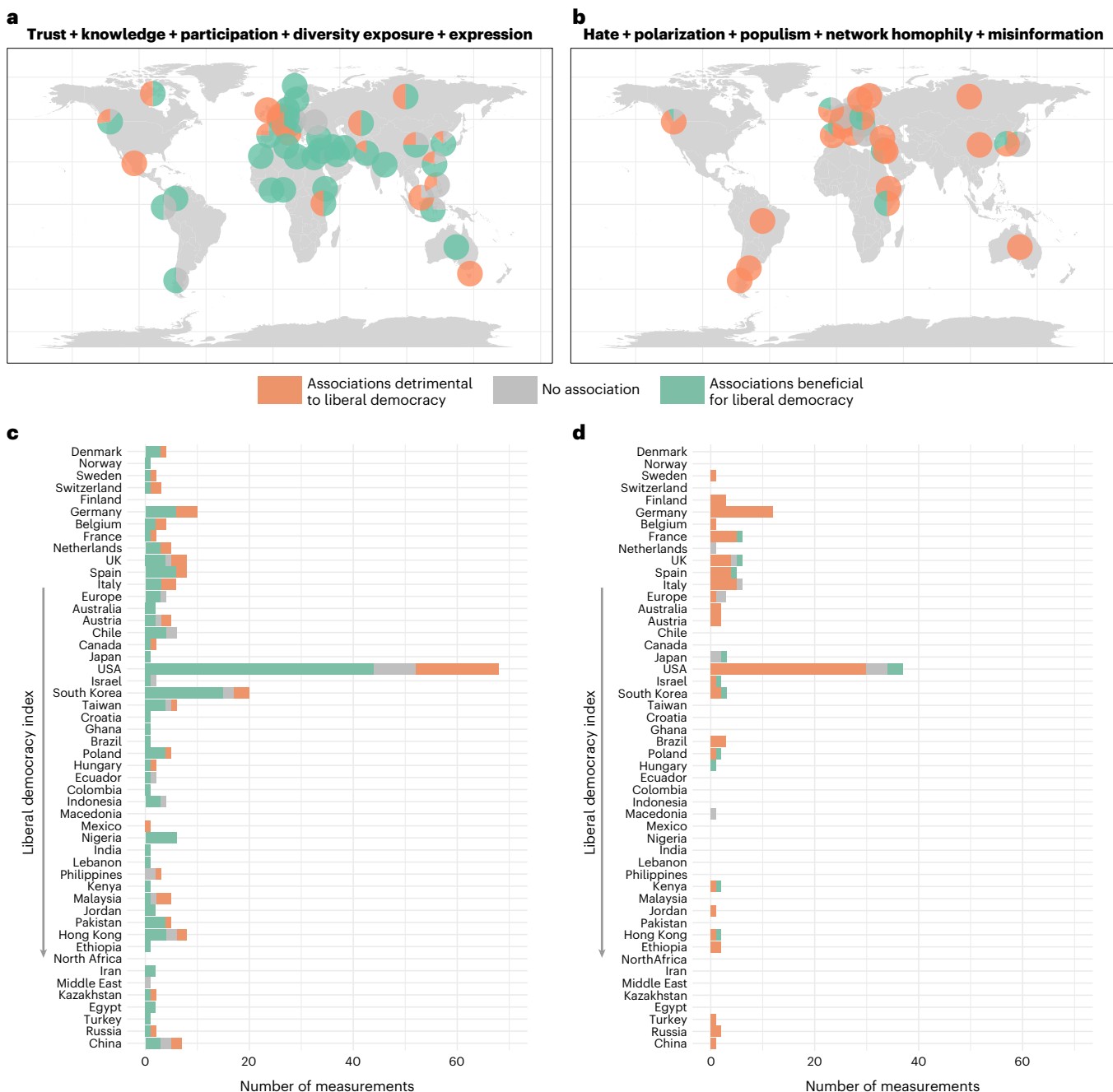

**Fig. 4 | Geographical distribution of associations showing beneficial and detrimental outcomes. a**, Geographical distribution of reported associations for the variables trust, knowledge, participation, exposure and expression. Pie charts show the composition of directions for each country studied. **b**, Geographic representation of reported associations for the variables hate, polarization, populism, homophily and misinformation. **c**, Data and variables in **a**, in absolute numbers of reported associations and sorted along the Liberal Democracy Index[18]. **d**, Data and variables in **b**, in absolute numbers of reported associations and sorted along the Liberal Democracy Index.

of actual behaviour on social media escaping the potential downsides of self-reporting. A potential blind spot in studies working with behavioural data from social media, inaccessible to both us and the original authors of the studies, is the selection of data provided by platforms. Therefore, it is tremendously important for researchers to get unrestricted access or, at least, transparent provision of random samples of data by platforms. The selection of users into the platforms, however, remains an open issue for behavioural data as it is often unclear who the active users are and why they are active online. We find that political outcome measures studied with behavioural data appear to show quite distinct results compared with those studied with large-scale

survey data. Combining both data types would probably maximize the chances for reliable conclusions about the impact of digital media on democracy.

We found relatively few null effects for some variables. This could be accurate, but it could also be driven by the file-drawer problem—the failure to publish null results. To examine the extent of a potential file-drawer problem, we contacted authors via large mailing lists but did not receive any unpublished work that fitted our study selection criteria. Regarding possible risk of bias, we found that only in 143 out of 354 measurements did authors clearly communicate that no conflict of interest was present (beyond the usual funding statement).

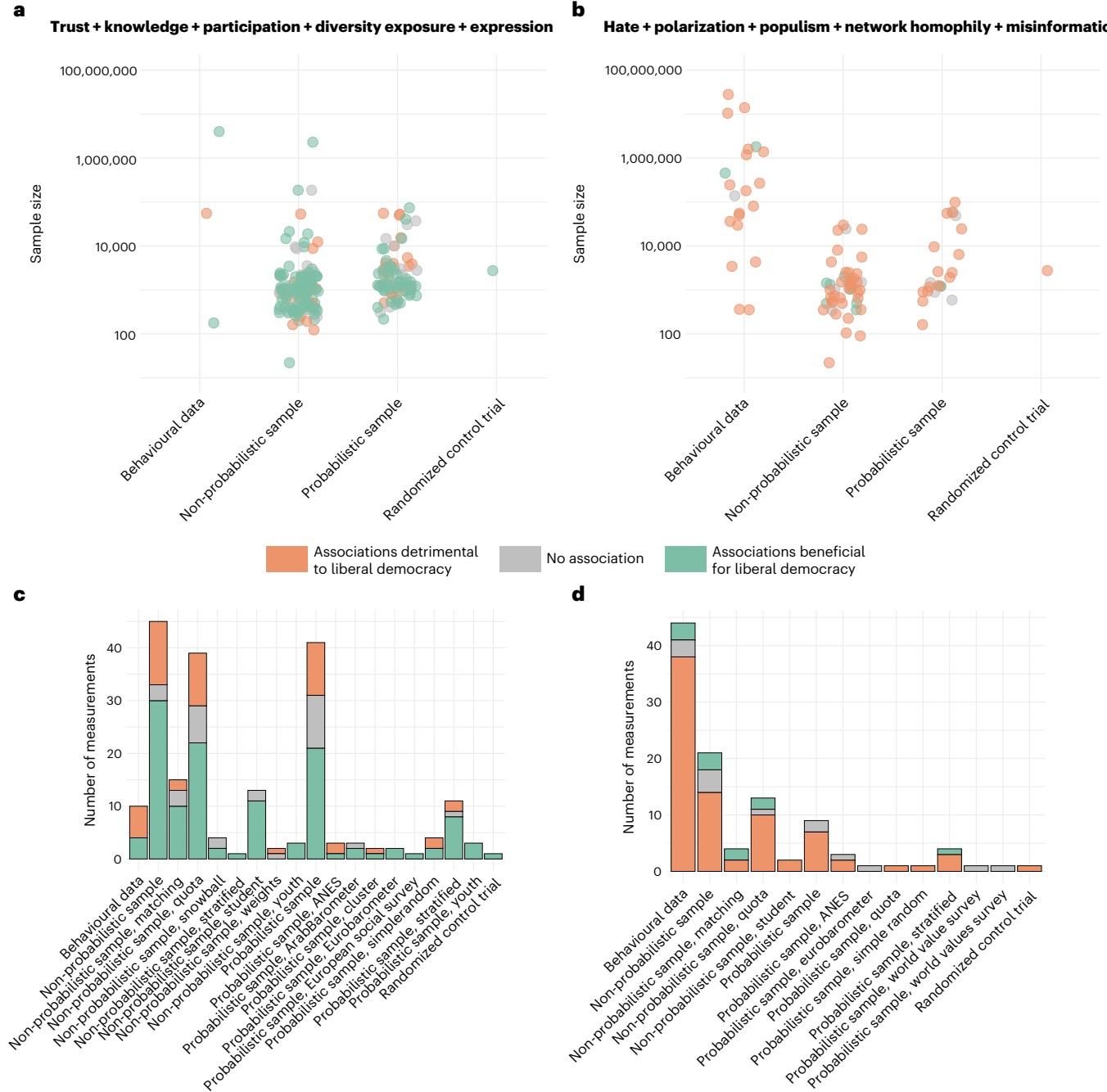

**Fig. 5 | Sample size and sampling strategy for reported associations.**
**a**, Sample size vs sampling methods for variables of trust, knowledge, participation, exposure and expression. Each dot represents one measurement, colour coded according to the direction of the reported association. **b**, Sample size vs sampling method for variables of hate, polarization, populims, network homophily and misinformation. **c**, More detailed breakdown for the same varibales as in **a** of sampling methods and their respective counts of reported associations and their direction. **d**, Breakdown of sampling methods and counts of associations for the same variables as in **b**.

However, we did not find a striking imbalance in the distribution of reported associations between those articles that did not explicitly state competing interest and those that did. Of the few associations for which conflicts of interest were stated, 4 pointed in beneficial, 3 in detrimental and 2 reported lack of directionality. In only 79 of 354 measurements did the researchers use open data practices. Considering articles that reported detrimental associations, we did not find a clear difference in the directions between those with and without open data. However, considering articles that reported beneficial outcomes, the numbers of positive findings in the studies without open data are relatively much larger than for the open science studies. Namely, 103 beneficial and 33 detrimental associations were reported in those without open data, while 19 beneficial versus 14 detrimental were reported in studies with open data practices. This observation might be due to the large number of survey-based studies about participation, which often do not follow open data practices. Even fewer of the studies in our sample were pre-registerd, namely, 13 of the 354, where 9 reported detrimental associations, only 3 reported beneficial associations and 1 found no direction of association. To shed light on other potential biases, we additionally examined temporal variations in

the directions of reported associations and found, besides the general explosive growth of studies in this domain, a slight trend towards an increasing number of both detrimental directions and null effects over time (Supplementary Fig. 4). At the author level, there was no clear pattern in the associations reported by those authors who published the greatest number of articles in our sample; several authors variously reported detrimental and beneficial effects as well as null effects, with a few exceptions (Supplementary Fig. 5). Their co-authorship network in Supplementary Fig. 6, split for the two types of outcomes measures, shows some communities of co-authors; however, no clear pattern of preferred direction of reported association can be spotted. Overall, we did not find evidence of a systematic bias in either direction driven by temporal trends or particular authors.

## Discussion

Regardless of whether they are authoritarian, illiberal, or democratic, governments around the world are concerned with how digital media affect governance and their citizenry's political beliefs and behaviours. A flurry of recent interdisciplinary research, stimulated in part by new methodological possibilities and data sources, has shed light on this potential interplay.

Although classical survey methods are still predominant, novel ways of linking data types, for example linking URL tracking data or social media data with surveys, permit more complex empirical designs and analyses. Furthermore, digital trace data allow an expansion in sample size. The articles we reviewed included surveys with a few hundred, up to a few thousand participants, but also large-scale social media analyses that included behavioural traces of millions. Yet with computational social science still in its early days, the amount of evidence supporting and justifying causal conclusions is still limited. Causal effects of digital media on political variables are also hard to pin down empirically due to a plethora of complexities and context factors, as well as the highly dynamic technological developments that make predicting the future difficult. While emergent political phenomena are hard to simulate in the lab, the value of estimation and data collection strategies to draw causal inferences from real-life data is enormous. However, the long-established trade-off between internal and external validity still applies, which also highlights the value of high-quality descriptive work.

Taking into account both correlational and causal evidence, our review suggests that digital media use is clearly associated with variables such as trust, participation and polarization. They are critical for the functioning of any political system, in particular democracies. Extant research reports relatively few null effects. However, the trends on each factor mostly converge, both across research methods and across correlative and causal evidence.

Our results also highlight that digital media are a double-edged sword, with both beneficial and detrimental effects on democracy. What is considered beneficial or detrimental will, at least partly, hinge on the political system in question: intensifying populism and network homophily may benefit a populist regime or a populist politician but undermine a pluralistic democracy. For democratic countries, evidence clearly indicates that digital media increase political participation. Less clear but still suggestive are the findings that digital media have positive effects on political knowledge and exposure to diverse viewpoints in news. On the negative side, however, digital media use is associated with eroding the 'glue that keeps democracies together'[33]: trust in political institutions. The results indicating this danger converge across methods. Furthermore, our results also suggest that digital media use is associated with increases in hate, populism and polarization. Again, the findings converge across causal and correlational articles.

Alongside the need for more causal evidence, we found several research gaps, including the relationship between trust and digital media and the seeming contradiction between network homophily and diverse news exposure. Methods that link tracking data for measuring news exposure with behavioural data from social media (for example, sharing activities or the sentiment of commenting) are crucial to a better understanding of this apparent contradiction.

## Limitations

The articles in our sample incorporate a plethora of methods and measures. As a result, it was necessary to classify variables and effects into broad categories. This is a trade-off we had to make in exchange for the breadth of our overview of the landscape of evidence across disciplines. For the same reason, we could not provide a quantitative comparison across the diverse sample of articles. We believe that digital media research would benefit from more unified measures (for example, for polarization), methods across disciplines to allow for better comparability in the future, a systematic comparison of different types of digital media (that is, Facebook and Twitter are neither of one kind nor, in all likelihood, are their effects) and extensions of outcome measurements beyond certain types of digital media. This follows other recent calls for commensurate measures of political and affective polarization[156]. The breadth of our review and the large number of political outcome measures in particular, made it necessary to be quite restrictive on other ends (see Fig. 6 for our exclusion process and Supplementary Table 1 for the detailed criteria). We explicitly decided to prioritize the selection of causal evidence (see Fig. 7 for an overview of the causal inference techniques that we considered) and other large-sample, quantitative, published evidence. However, following this pre-registered search strategy led to the selection of unequal numbers of studies for different outcome variables. For example, our search query selected considerably more studies examining political participation than political expression or trust, while at the same time, it did not include all studies that are included in other systematic reviews[45] due to stricter exclusion criteria.

The interpretation of our results was in several cases hampered by a lack of appropriate baseline measures. There is no clear measure of what constitutes a reasonable benchmark of desirable political behaviour in a healthy democracy. In addition, there were no means of quantification of some of these behaviours in the past, outside of digital media. This problem is particularly pronounced for factors such as exposure to diverse news, social network homophily, misinformation and hate speech. Measuring these phenomena at scale is possible through digital media (for example, by analysing social network structure); much less is known about their prevalence and dynamics in offline settings. Many articles therefore lacked a baseline. For instance, it is neither clear what level of homophily in social networks is desirable or undesirable in a democratic society, nor is it clear how to interpret the results of certain studies on polarization[69,130], whose findings depend on whether one assumes that social media have increased or decreased exposure to opposing views relative to some offline benchmark. For example, if exposure to opposing views is increased on social media, the conclusion of one study[130] would be that it reduces polarization, but if exposure is decreased, one would come to the opposite conclusion. Notably, in this study, counter-attitudinal exposure was found to be down-ranked by Facebook's news feed—hence supporting a process that fosters polarization instead of counteracting it. Furthermore, results about populism might be skewed: descriptive evidence on the relative activity and popularity of right-wing populist parties in Europe suggests their over-representation, as in the case of Germany's AfD, on social media, relative to established democratic parties (see, for example, ref. [132]). Therefore, it is difficult to interpret even causal effects of digital media use on populist support in isolation from the relative preponderance of right-wing content online.

## Conclusion

Our results provide grounds for concern. Alongside the positive effects of digital media for democracy, there is clear evidence of serious threats to democracy. Considering the importance of these corrosive and potentially difficult-to-reverse effects for democracy, a

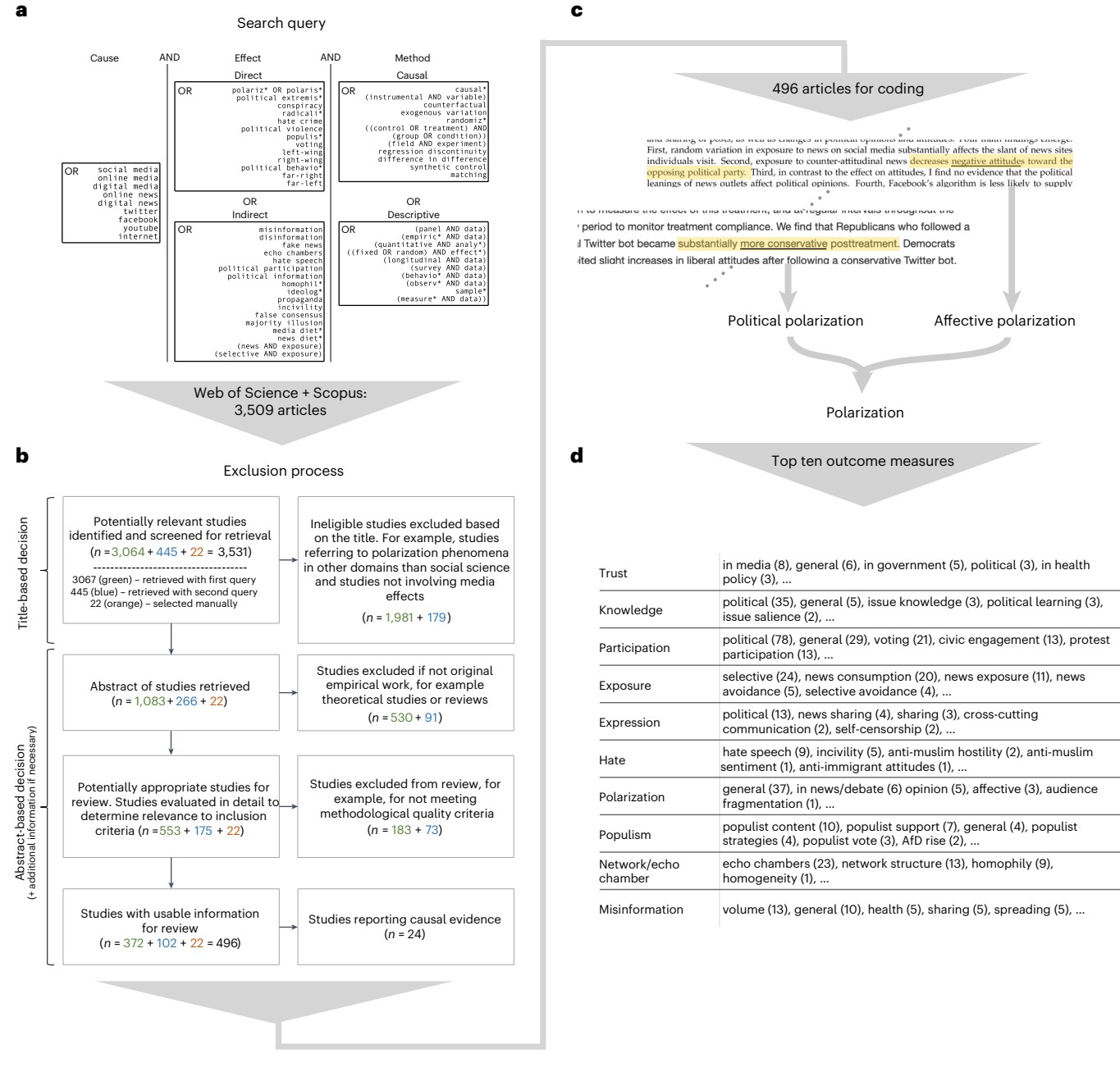

**Fig. 6 | Strategy for curating the sample of relevant articles. a**, Keywords included in our search query, run on Web of Science and Scopus, with logical connectors. Focus was on causal inference methods (method column), but also inclusion of descriptive quantitative evidence, relationships between digital media (cause column) and political outcomes (direct effect box) or content features (indirect effect box). **b**, Flowchart representing the stepwise exclusion process, starting with title-based exclusion, followed by abstract-based exclusion. **c**, Example illustration of outcome variable extraction from the abstracts. **d**, Breakdown of the most frequently reported political variables into top 10 categories. Numbers in brackets are counts of measurements in the set.

better understanding of the diverging effects of digital media in different political contexts (for example, authoritarian vs democratic) is urgently needed. To this end, methodological innovation is required. This includes, for instance, more research using causal inference methodologies, as well as research that examines digital media use across multiple and interdependent measures of political behaviour. More research and better study designs will, however, also depend on access to data collected by the platforms. This access has been restricted or foreclosed. Yet without independent research that has unhampered access to all relevant data, the effects of digital media can hardly be understood in time. This is even more concerning because digital media

can implement architectural changes that, even if seemingly small, can scale up to widespread behavioural effects. Regulation may be required to facilitate this access[157]. Most importantly, we suggest that the bulk of empirical findings summarized here can be attributed to the current status quo of an information ecosystem produced and curated by large, commercial platforms. They have succeeded in attracting a vast global audience of users. The sheer size of their audience as well as their power over what content and how content gets the most attention has led, in the words of the philosopher Jürgen Habermas, to a new structural transformation of the public sphere[16]. In this new public sphere, everybody can be a potential author spontaneously producing

**Fundamental Principle of Causal Inference**

In experiments with randomized treatment allocation and perfect compliance, the difference between treatment and control groups can be interpreted as the causal average treatment effect (ATE), not considering measurement error.
However, in observational settings or imperfect experiments, the identification of ATE is likely impeded by the existence of confounders and/or colliders.
Causal inference techniques aim to get as close as possible to the ideal experimental standard using various statistical strategies, such as:

| Matching | Instrumental variables | Panel designs |
|---|---|---|
| **Explicitly balancing treatment and control units.** | **Isolating exogenous variation in the treatment variable to induce as-if randomizaton.** | **Partialling out observed and unobserved unit and/or time invariant confounders.** |

**Matching**

The goal is to compare units that are similar in all respects but the treatment (e.g. exposure to social media). In principle, one could split the sample into strata of potential confounders and compare cases between these strata. Various techniques such as

- **Nearest neighbor covariate matching**
- **Propensity score matching**
- **Coarsened exact matching**

make it possible to overcome the 'curse of dimensionality' that comes with many possible confounding variables and not enough *exact* matches in the control group for every unit in the treatment group by finding the **most plausible** counterfactual in the control group.

**Instrumental variables**

By splitting the variation of the treatment variable (e.g. internet use) in two parts—one potentially related to confounders and one truly exogenous that is caused by other factors (instrumental variables, IV) that are unrelated to confounders (exclusion restriction)—one can identify the partial (local) causal average treatment effect (LATE). IVs can be used in observational settings or in (field) experiments with imperfect compliance.

**Two-stage-least-squares (2SLS) strategies**
1. Regress the treatment status on the instrumental variable (or the treatment assignment in experimental settings)
2. Regress the outcome on the predicted values of the treatment

**Panel designs**

The combination of cross-sectional and time-series data in panel settings allows the comparison of **changes between** treatment and control units over time (e.g. in fixed-effects or difference-in-difference designs), which relaxes certain assumptions and allows the consideration of **unobserved** confounders.

For example in least squares dummy variable regressions (LSDV)

- **Time fixed effects** control for unit-specific, time-invariant confounders
- **Unit fixed effects** control for time varying global variables that are the same for both unit and control units

*X = treatment/independent variable, Y = outcome/dependent variable, C = measured confounder, U/V = unobserved confounders, M = matching variable (e.g. propensity score), I = instrumental variable, G = unit, T = time.*

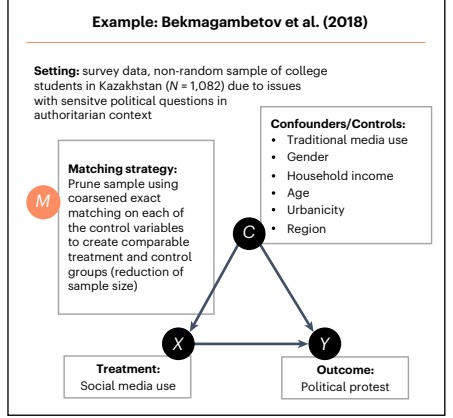

**Example: Bekmagambetov et al. (2018)**

**Setting:** survey data, non-random sample of college students in Kazakhstan (N = 1,082) due to issues with sensitve political questions in authoritarian context

**Matching strategy:** Prune sample using coarsened exact matching on each of the control variables to create comparable treatment and control groups (reduction of sample size)

**Confounders/Controls:**
- Traditional media use
- Gender
- Household income
- Age
- Urbanicity
- Region

**Treatment:** Social media use

**Outcome:** Political protest

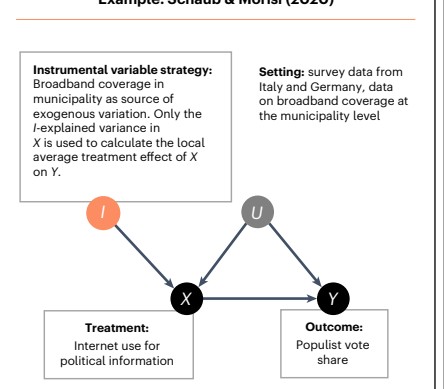

**Example: Schaub & Morisi (2020)**

**Instrumental variable strategy:** Broadband coverage in municipality as source of exogenous variation. Only the *I*-explained variance in *X* is used to calculate the local average treatment effect of *X* on *Y*.

**Setting:** survey data from Italy and Germany, data on broadband coverage at the municipality level

**Treatment:** Internet use for political information

**Outcome:** Populist vote share

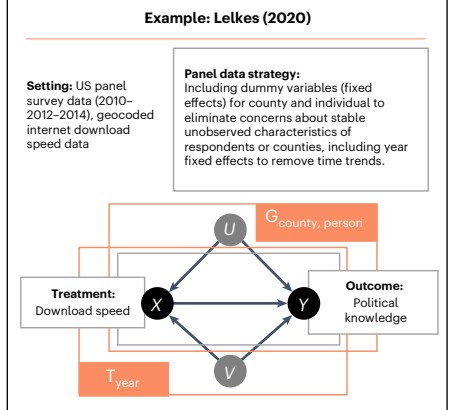

**Example: Lelkes (2020)**

**Setting:** US panel survey data (2010–2012–2014), geocoded internet download speed data

**Panel data strategy:** Including dummy variables (fixed effects) for county and individual to eliminate concerns about stable unobserved characteristics of respondents or counties, including year fixed effects to remove time trends.

**Treatment:** Download speed

**Outcome:** Political knowledge

**Fig. 7 | Summary of causal inference techniques used.** Fundamental principles of causal inference techniques and statistical strategies used in our sample of causal evidence (excluding field experiments).

content, both right-wing radical networks as well as the courageous Belarusian women standing up for human rights and against a repressive regime. One need not share Habermas' conception of 'deliberative democracy' to see that current platforms fail to produce an information ecosystem that empowers citizens to make political choices that are as rationally motivated as possible. Our results show how this ecosystem plays out to have important consequences for political behaviours and attitudes. They further underscore that finding out which aspects of this relationship are detrimental to democracy and how they can be contained while actively preserving and fostering the emancipatory potential of digital media is, perhaps, one of the most important global tasks of the present. Our analysis hopes to contribute to the empirical basis of this endeavour.

## Methods

This systematic review follows the MOOSE Guidelines for Meta-Analyses and Systematic Reviews of Observational Studies[158]. The detailed protocol of the review process was pre-registered on the Open Science Framework (OSF) at https://osf.io/7ry4a/. The repository also contains the completed MOOSE checklist showing where each guideline is addressed in the text.

Figure 6 summarizes the search query that we used on two established academic databases, Scopus and Web of Science (both highly recommended search tools), the resulting number of articles from the query and the subsequent exclusion steps, leading to the final sample size of N = 496 articles under consideration[159–575].

### Study selection criteria

We included only original, empirical work. Conceptual or theoretical work, simulation studies and evidence synthesizing studies were excluded. Articles had to be published in academic journals in English. Unpublished studies for which only the abstract or a preprinted version was available were excluded from the review. We excluded small-*N* laboratory experiments and small-*N* student surveys (*N* < 100) from our body of original work due to validity concerns. Although correlational evidence cannot establish a causal direction, we focused on articles that

examined effects of digital media on democracy but not the opposite. We therefore excluded, for example, articles that examined ways to digitize democratic procedures. To be included, articles had to include at least two distinct variables, a digital media variable and a political outcome. Articles measuring a single variable were only included if this variable was a feature of digital media (for example, hate speech prevalence, homophily in online social networks, prevalence of misinformation in digital media).

### Search strategy, study selection, coding and data extraction

Articles eligible for our study had to be published before 15 September 2021. We sourced our review database from Scopus and Web of Science, as suggested by ref. [159]. The search query (Fig. 6) was constructed in consultation with professional librarians and was designed to be as broad as possible to pick up any articles containing original empirical evidence of direct or indirect effects of digital media on democracy (including correlational evidence). We further consulted recent, existing review articles in the field[32,39,40] to check for important articles that did not appear in the review body. Articles that were included manually are referenced separately in the flowchart (Fig. 6). In addition, we contacted authors via large mailing lists of researchers working on computational social science and misinformation but did not receive any unpublished work that fitted our study selection criteria. The query retrieved $N = 3,509$ articles. Of these, 1,349 were retained after screening the titles for irrelevant topics. This first coding round, whether an article, based on the title, fits the review frame or not, was split between two coders. Coders could flag articles that are subject to discussion to let the other coder double check the decision. In this round, only clearly not fitting articles were excluded from the sample. A list of exclusion criteria can be found in Supplementary Information.

The next coding round, whether an article, based on the abstract, fits the review frame, was conducted in parallel by two coders. The inter-coder reliability, after this round of article selection, was Krippendorff's alpha of 0.66 (87% agreement). After calculating this value, disagreement between coders was solved through discussion. At this stage, we excluded all studies that were not original empirical work, such as other reviews or conceptual articles, simulation studies and purely methodological articles (for example, hate speech or misinformation detection approaches). This coding round was followed by a more in-depth coding round. Here we refined our exclusion decisions; for example, we excluded studies that examined the digitization of government, preprints, small-scale lab experiments, small-scale convenience or student samples and studies that only included one variable (for example, description of online forums) (see Supplementary Table 1 for a detailed list of criteria). A full-text screen was performed in cases where the relevant information could not be retrieved from the abstract and for all articles implying causal evidence.

After both rounds of abstract screening, 474 articles remained in our sample. After cross-checking the results of our literature search against the references from existing reviews, we found and included further $N = 22$ articles that met our thematic criteria but were not identified by our search string. Ultimately, a total of 496 articles were selected into the final review sample. Figure 6b summarizes the selection procedure.

The following information was extracted from each article using a standardized data extraction form: variable groups under research (digital media, features of media and/or political outcome variables), the concrete digital media under research, the explicit political outcome variable, the methods used, the country of origin, causal claims, possible effect heterogeneity (moderation) as well as various potential sources of bias. To assess various quality criteria of the studies, the coders had to visit the full text of the articles (for example, to find the declaration of competing interests, pre-registration or data availability statements, or to consider the methods section). Therefore, and facing the large number of articles under consideration, blinding could not be established during this procedure.

When conducting a systematic review with a broad scope, categories of the variables cannot be exhaustively defined before coding. Therefore, variable categories, especially for the digital media variables and the political outcome variables, were chosen inductively. In the first extraction step, coders stuck closely to the phrasing of the authors of the respective study. To reduce redundancy and refine the clustering of the variables, we iteratively generated frequency tables and manually sorted single variables to the best-fitting categories until a small number of clearly distinct categories was selected. After the categories were defined, both coders re-coded 10% of the sample to calculate inter-coder reliabilities for all key variables. We provide a table of inter-coder reliabilities (percentage agreements and Krippendorff's alphas) (Supplementary Table 2).

### Data synthesis and analysis

Due to considerable heterogeneity in methods in the articles—including self-report surveys through network analysis of social media data, URL tracking data and field experiments—no calculation of meta-analytic effect sizes was possible. The final table of selected articles with coded variables will be published alongside this article as a major result of this review project. The effect directions of 10 important political outcome variables (4 consistent with liberal democracy, 4 opposing democratic values) are summarized in Fig. 2. For articles dealing with these political variables, we also assessed the country in which the study was conducted (Fig. 4), as well as explicit sources of effect heterogeneity such as demographic characteristics of study participants or characteristics of the digital media platform.

For the overview analysis, which includes both correlational and causal evidence, we mainly restricted ourselves to the evaluation effects reported in the abstracts. Articles making explicit causal claims and/or using causal inference methods (Fig. 7) were examined in-depth and summarized as simplified path diagrams with information on mediators, moderators, country of origin and method used (Fig. 3).

### Deviations from the protocol

The volume of papers our query returned prevented an in-depth analysis of confounding variables. Instead, our assessment of quality relied on the sampling strategy and sample size, the method used, sources of heterogeneity and transparency criteria, such as open data practices and pre-registration. Furthermore, we were able to construct the co-author network by matching the author's names, but were unable to produce a meaningful co-citation network due to the incompleteness and ambiguity of references in the export format that we used.

### Reporting summary

Further information on research design is available in the Nature Research Reporting Summary linked to this article.

## Data availability

The dataset including all originally collected studies with decision stages ($N = 3,531$, 'full_data.xlsx'), the table including all papers within our review sample ($N = 496$, 'data_review.xlsx') and the table including all effects reported within papers dealing with the top ten outcome measures ($N = 354$, 'data_effects.xlsx') are available at https://osf.io/7ry4a/.

## Code availability

R scripts for all analyses and figures are available at https://osf.io/7ry4a/.

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

## Acknowledgements

We thank S. Munzert for providing his perspective on causal inference and issues specific to political science, D. Ain for editing the manuscript and F. Stock for help in the literature comparison. P.L.-S., S.L. and R.H. acknowledge financial support from the Volkswagen Foundation (grant 'Reclaiming individual autonomy and democratic discourse online: How to rebalance human and algorithmic decision-making'). S.L. acknowledges support from the Humboldt Foundation through a research award and partial support by an ERC Advanced Grant (PRODEMINFO) during completion of this paper. L.O. acknowledges financial support by the German National Academic Foundation in the form of a PhD scholarship. The authors received no specific funding for this work. The funders had no role in study design, data collection and analysis, decision to publish or preparation of the manuscript.

## Author contributions

All authors designed the study. P.L.-S and L.O. selected and coded the literature. P.L.-S. and L.O. evaluated the coded tables. All authors analysed the results and wrote the manuscript.

## Funding

## Competing interests

The authors declare no competing interests.

## Additional information

**Correspondence and requests for materials** should be addressed to Philipp Lorenz-Spreen.

# Reporting Summary

## Statistics

For all statistical analyses, confirm that the following items are present in the figure legend, table legend, main text, or Methods section.

| n/a | Confirmed | |
|---|---|---|
| ☐ | ☒ | The exact sample size ($n$) for each experimental group/condition, given as a discrete number and unit of measurement |
| ☒ | ☐ | A statement on whether measurements were taken from distinct samples or whether the same sample was measured repeatedly |
| ☒ | ☐ | The statistical test(s) used AND whether they are one- or two-sided _Only common tests should be described solely by name; describe more complex techniques in the Methods section._ |
| ☒ | ☐ | A description of all covariates tested |
| ☒ | ☐ | A description of any assumptions or corrections, such as tests of normality and adjustment for multiple comparisons |
| ☒ | ☐ | A full description of the statistical parameters including central tendency (e.g. means) or other basic estimates (e.g. regression coefficient) AND variation (e.g. standard deviation) or associated estimates of uncertainty (e.g. confidence intervals) |
| ☒ | ☐ | For null hypothesis testing, the test statistic (e.g. $F$, $t$, $r$) with confidence intervals, effect sizes, degrees of freedom and $P$ value noted _Give P values as exact values whenever suitable._ |
| ☒ | ☐ | For Bayesian analysis, information on the choice of priors and Markov chain Monte Carlo settings |
| ☒ | ☐ | For hierarchical and complex designs, identification of the appropriate level for tests and full reporting of outcomes |
| ☒ | ☐ | Estimates of effect sizes (e.g. Cohen's $d$, Pearson's $r$), indicating how they were calculated |

_Our web collection on statistics for biologists contains articles on many of the points above._

## Software and code

Policy information about availability of computer code

| | |
|---|---|
| Data collection | Scopus (September, 2021), Web-of-Science (September, 2021) |
| Data analysis | R version 4.1.2 |

For manuscripts utilizing custom algorithms or software that are central to the research but not yet described in published literature, software must be made available to editors and reviewers. We strongly encourage code deposition in a community repository (e.g. GitHub). See the Nature Portfolio guidelines for submitting code & software for further information.

## Data

Policy information about availability of data

All manuscripts must include a data availability statement. This statement should provide the following information, where applicable:
- Accession codes, unique identifiers, or web links for publicly available datasets
- A description of any restrictions on data availability
- For clinical datasets or third party data, please ensure that the statement adheres to our policy

The fully coded table of articles as well as scripts for all figures are available at: https://osf.io/7ry4a/.

# Field-specific reporting

Please select the one below that is the best fit for your research. If you are not sure, read the appropriate sections before making your selection.

☐ Life sciences   ☒ Behavioural & social sciences   ☐ Ecological, evolutionary & environmental sciences

For a reference copy of the document with all sections, see nature.com/documents/nr-reporting-summary-flat.pdf

# Behavioural & social sciences study design

All studies must disclose on these points even when the disclosure is negative.

| | |
|---|---|
| Study description | Systematic review, no primary human participant data was collected |
| Research sample | Causal and correlational studies examining digital media and political variables |
| Sampling strategy | - |
| Data collection | Studies for our review were collected using a systematic search query (provided in the appendix) on Scopus and Web-of-Science. The researchers were not blinded. |
| Timing | The latest search query run was conducted in September 2021, no gaps in the sample exist as this query includes all studies that were published before this date. |
| Data exclusions | We did not include conceptual or theoretical studies, small-N experiments or small-N survey, simulation studies and other evidence synthesizing work (exclusion criteria are described in the materials and methods section). Those exclusion criteria were pre-registered. |
| Non-participation | No participants were included in our study. |
| Randomization | For a systematic review, no primary data is collected, hence no experimental groups. |

# Reporting for specific materials, systems and methods

We require information from authors about some types of materials, experimental systems and methods used in many studies. Here, indicate whether each material, system or method listed is relevant to your study. If you are not sure if a list item applies to your research, read the appropriate section before selecting a response.

## Materials & experimental systems

| n/a | Involved in the study |
|---|---|
| ☒ | ☐ Antibodies |
| ☒ | ☐ Eukaryotic cell lines |
| ☒ | ☐ Palaeontology and archaeology |
| ☒ | ☐ Animals and other organisms |
| ☒ | ☐ Human research participants |
| ☒ | ☐ Clinical data |
| ☒ | ☐ Dual use research of concern |

## Methods

| n/a | Involved in the study |
|---|---|
| ☒ | ☐ ChIP-seq |
| ☒ | ☐ Flow cytometry |
| ☒ | ☐ MRI-based neuroimaging |

