## [Peer Review File · Nature Human Behaviour]

Peer Review Information

Journal: Nature Human Behaviour

Manuscript Title: A Systematic Review of Worldwide Causal and Correlational Evidence on Digital Media and Democracy

Corresponding author name(s): Philipp Lorenz-Spreen

Reviewer Comments & Decisions:

Decision Letter, initial version:

16th February 2022

Dear Dr. Lorenz-Spreen,

Thank you once again for your manuscript, entitled "Digital Media and Democracy: A Systematic Review of Causal and Correlational Evidence Worldwide", and for your patience during the peer review process.

Your Article has now been evaluated by 3 referees. You will see from their comments copied below that, although they find your work of potential interest, they have raised quite substantial concerns. In light of these comments, we cannot accept the manuscript for publication, but would be interested in considering a revised version if you are willing and able to fully address reviewer and editorial concerns.

We hope you will find the referees' comments useful as you decide how to proceed. If you wish to submit a substantially revised manuscript, please bear in mind that we will be reluctant to approach the referees again in the absence of major revisions. We are committed to providing a fair and constructive peer-review process. Do not hesitate to contact us if there are specific requests from the reviewers that you believe are technically impossible or unlikely to yield a meaningful outcome.

To guide the scope of the revisions, the editors discuss the referee reports in detail within the team, including with the chief editor, with a view to (1) identifying key priorities that should be addressed in revision and (2) overruling referee requests that are deemed beyond the scope of the current study. We hope that you will find the prioritised set of referee points to be useful when revising your study. Please do not hesitate to get in touch if you would like to discuss these issues further.

In particular, your revision must address the following (as well as all other reviewer comments):

1) Address concerns raised by Reviewers 2 and 3 regarding the conceptualization of your independent variable,

ensuring that your inclusion criteria are fully justified and that the limitations of your approach are transparently considered.

- 2) Ensure that your study follows your pre-registration, with any necessary deviations thoroughly explained and justified, and any exploratory analyses clearly labeled as such.
- 3) Address reviewer concerns regarding interpretations and conclusions, including about publication bias and the file drawer problem.
- 4) Provide the methodological information requested by Reviewer 1, as well as a completed MOOSE checklist.
- 5) With respect to Reviewer 1's concerns regarding the clarity of your research question, we ask that you do not alter the question itself (it should remain consistent with the original preregistration). However, we recommend that you add a discussion of the different components of the question, to make it easier for readers to quickly grasp the objectives of the work, as suggested by Reviewer 1.

If you wish to submit a suitably revised manuscript we would hope to receive it within 4 months. We understand that the COVID-19 pandemic is causing significant disruptions which may prevent you from carrying out the additional work required for resubmission of your manuscript within this timeframe. If you are unable to submit your revised manuscript within 4 months, please let us know. We will be happy to extend the submission date to enable you to complete your work on the revision.

- Include a "Response to the editors and reviewers" document detailing, point-by-point, how you addressed each editor and referee comment. If no action was taken to address a point, you must provide a compelling argument. This response will be used by the editors to evaluate your revision and sent back to the reviewers along with the revised manuscript.
- Highlight all changes made to your manuscript or provide us with a version that tracks changes.

[REDACTED]

Thank you for the opportunity to review your work. Please do not hesitate to contact me if you have any questions or would like to discuss the required revisions further.

Sincerely,
Aisha

Aisha Bradshaw, PhD
Senior Editor
Nature Human Behaviour

Reviewer expertise:

Reviewer #1: epidemiology, systematic reviews and evidence synthesis methods

Reviewer #2: political communication

Reviewer #3: political communication

REVIEWER COMMENTS:

Reviewer #1:
Remarks to the Author:

This systematic review addresses a timely and important question: what is the relationship between digital technology and democratic processes? This is likely to be of interest to a wide audience of readers. A large number of articles (k=498) were included, most of which were correlational, but a small subset (k=21) used causal inference methods to enable stronger conclusions to be drawn. Below I provide comments and suggestions which predominantly focus on improving the methodological aspects of the study.

1. The authors should report the MOOSE guidelines checklist including page numbers to show where each guideline has been addressed in the manuscript. Without the checklist, it is not clear that these guidelines have been addressed.

2. The pre-registration stated that the authors will assess study quality and sources of bias. Was this assessed and could the authors evaluate the evidence in line with the quality of the studies, and sources of bias?

3. The pre-registration stated that the authors will conduct and report a network analysis among the co-authors as well as co-citations of included papers. I could not see this in the systematic review. If the authors decided not to include this, they should explain why.

4. Study selection criteria: how were small-n lab experiments and small-N student surveys defined? I.e., what was the minimum N?

5. Study selection criteria: The inclusion criteria statement ("To be included, articles had to include at least two distinct variables, treatment or independent variable and outcome") wasn't particularly specific. I recommend the authors omit the terms "treatment variable", "independent variable" and "outcome" and replace them with the actual

variables of interest.

6. Discussion: it seems unrealistic to suggest that publication bias is unlikely to exist because null effects of digital media on political outcomes would be important enough to report. While I agree that the null effects would be important, this is also the case for many other research questions where publication bias has been evidenced to exist. To examine potential publication bias / the file drawer problem, could the authors identify any pre-registered articles included in this review to see if they were more likely to report null results? And/or perhaps consult other quantitative meta-analyses on this or similar topics to see if they have identified publication bias?

7. The authors should be commended for providing their dataset and figure code on the OSF. Two follow-up points on this: (1) Could the authors please state the name of the file that contains the dataset? E.g., is it "Data_effects_complete" or "data_final"? (2) It is stated that "code for the analyses and visualisations" is provided on the OSF, but only code used to derive the Figures is on the OSF. Was there any other code used to analyse data for the article? If so, could that be provided?

8. The pre-registered question "If, to what degree and in which contexts, do digital media have detrimental effects on democracy?" seemed to be phrased in a complex way and include multiple components. I am not sure if it would be appropriate to edit it, given it was pre-registered, but if there is scope to do so, the authors could consider breaking it down into multiple questions, e.g: (1) "To what extent does digital media have detrimental effects on democracy?", and (2) "In what contexts does digital media have detrimental effects on democracy?".

9. Data synthesis and analysis: The sentence "The final table of selected articles with coded variables will be published alongside this article as a major result of this review project" should be updated to provide the link to the dataset of selected articles with variables.

10. Figure 2. For the insets, it is difficult to understand the labels – e.g., do they span multiple lines or is it one label per line? Also on the left hand side inset, there seem to be some labels with no data for either top or bottom panel – do they need to be included? Could these insets be stretched to show the labels more clearly? Or put as a separate figure to avoid detracting attention from the middle part, which is much easier to understand?

11. Figure 3. Three points: (1) Field experiments seem to be a common causal inference strategy in the included studies, but field experiments were not included in Fig 3, and it is unclear what they involved (from Fig. 4). I suggest adding a description of Field experiments to Fig 3. (2) There are many causal inference strategies other than regression discontinuity that are not shown in the figure (e.g., twin studies, Mendelian randomisation, natural experiments), so perhaps rephrase the note to acknowledge this or omit it? (3) There is a small typo ("treatment") in the "Matching" section.

12. Figure 4. Some outcome terms were not clear, e.g. "correct voting" (what qualifies as correct?), "voting congruency" (congruent with what?), "turnout" (does this mean voter turnout?). I recommend the authors provide a table or further information defining the outcome terms.

13. Figure 5. Fig 1b implies that 3-10 articles were from the UK, but the UK is not shown in Fig 5 – why was this?

14. Fig. 6: Could the authors make it clearer which articles were referenced manually and were not identified in the original search?

Reviewer #2:

Remarks to the Author:

The manuscript "Digital Media and Democracy: A Systematic Review of Causal and Correlational Evidence Worldwide" provides a large-scale systematic review (N = 498 articles) of research examining the role of "digital media" and its influences on various political outcome variables.

Key results: Previous research in the field has examined effects on specific political variables like, for instance, political polarization, political participation, various forms of trust and political knowledge. This is the first review of its kind (to the best of my knowledge) to systematically review various political outcome variables together in an integrated manner. This manuscript substantially broadens our understanding regarding the effects of digital media. Examining political outcome variables in an integrated way can be regarded beneficial, as effects of digital media are complex affecting very different processes (at the same time), which may be deemed (from a normative perspective) both positive (i.e., political participation) and negative (i.e., political polarization). Looking at different outcome variables separately can certainly be informative. Yet, this review provides a broader perspective by examining very different outcome variables in an integrated way. Also, it can help to disentangle different processes and clarify previous findings in the field, as the authors systematically differentiate between correlational and causal effects of digital media.

The results show both beneficial (e.g., effects on political participation) and detrimental effects (e.g., political polarization) for democracy and thus "ground for concern" (p. 16), as there is "clear evidence of serious threats to democracy" (p. 16), as the authors put it.

Validity: The manuscript follows a well-planned, transparent (i.e., preregistered) and well-executed procedure. First, one concern I have relates to the authors' understanding of "digital media", which is not explicitly defined and operationalized. This should be changed. A clear definition should be provided early on in the manuscript, as the concept of "digital media" is obviously of key importance for the paper. Importantly, differences between "digital media" (in the authors' understanding) and other types of media should be explained. On page 2 there is a hint. Authors may regard platforms that "do not create content" themselves as "digital media". But this needs more explanation and justification. Also, on page 4 it is highlighted that authors do "not consider the effects of traditional media (e.g., television or radio)". However, today such "traditional media contents" are regularly available online or on social media (i.e., "digital media"), be it on demand on platforms of traditional media outlets or online newspaper content that is spread via social media. In fact, a large proportion of "news" and links on social media stems from "traditional media outlets", which move their content online (TV news available on YouTube; footage, links etc. can be shared on Twitter), radio programs that originally aired on national radio are then also available as a podcast on a commercial platform etc. All this being said, I would like to know how exactly the authors define "digital media" and how their understanding may have influenced the search process for articles and, in turn, the results. Would a different conceptualization of digital media potentially lead to different results?

I would like to emphasize that I do NOT regard the missing of a clear definition as a fundamental flaw or major problem that cannot be fixed. Instead, I would like to encourage authors to think about this aspect more, provide answers to my questions, and to revise the manuscript accordingly.

Second, I have some questions and concerns regarding the article selection process and its reliability. The search-query presented in Appendix A.1. seems to be well-developed and appropriate. My first question relates to the article exclusion procedure. Can the authors describe this process on page 18 a bit more? The number of articles was narrowed down from 3,518 to 1,351 to 741 and finally to 498. Authors provide some information on this procedure in the Materials and Methods section, but to me it remains unclear if coders/raters mutually agreed on study exclusion in

these different steps. Did the authors measure this in any way? Do interrater-coefficients show acceptable results in regard to these first study exclusion steps?

Third, one concern I have relates to the rather low Krippendorff's alpha value of 0.66 reported on page 18. In the literature values in this range are rather critically discussed. Of course, it always depends on the particular research context/problem at hand meaning that coefficients have to be "put into study context". Anyways, I would like to see a more critical discussion on this aspect of intercoder reliability and how it relates to your findings. What are potential limitations here? Also, I am not quite sure how your "overall" score of 87% relates to the Krippendorff and Cohen coefficients reported. Maybe, you can briefly explain this aspect. Finally, I would like to see a more fine-grained explanation of reliability coefficients in regard to the major categories that were coded, as reliability may be better in some categories compared to others and the overall reliability scores are thus not perfectly informative. Maybe authors can provide some more information and, for instance, insert a table (overview of reliability coefficients in regard to different categories).

Originality and significance:

Based on an extensive review of research some of the key claims of the present manuscript are that there are both positive and negative effects of digital media on different political outcome variables. Six key areas of influence are identified: participation, trust, political knowledge, polarization, populism, network structures and news exposure.

Also, the authors see "clear evidence of serious threats to democracy". The findings suggest that much more research needs to study the role of "digital media" in other (non-Western) political contexts (e.g., authoritarian vs. democratic) and authors see a need for more research that examines causal effects of digital media on political outcome variables, as a lot of research in the field follows correlational approaches.

This paper significantly contributes to our understanding of digital media and effects on political behavior. The results are relevant and of interest to researchers and students in different disciplines (communication science, political science, psychology and intersecting fields such as political psychology) and many others interested in the question of how digital media affects politics and political behavior.

This being said, I suggest that authors explain the paper's purpose and innovation in an even better way. That is, "Why did the authors decide for this very broad methodological approach"? I can see the point and I generally welcome including multiple outcome variables to get an overall picture. Yet, at the same time this should be better explained and "justified", as there are also some downsides to this kind of approach (that should be explicitly named), which can be criticized. Why is one "overall review" better/needed (closes an important research gap?) compared to more focused reviews that focus on just one key dependent variable? I think, this needs to be much better justified and explained early on. Maybe authors can think about slightly re-structuring the sections on page 3/4 and say something about existing reviews and meta-analyses, why the present paper is important (why we need it), how it closes a key research gap and if other papers in the field followed a similar approach (or if the present paper is the first one in doing so).

Overall, the paper presents an innovative and novel methodological approach and reports results that are important and relevant to human behavior.

Data & methodology:

Authors present a valid and transparent approach of data selection and analysis. Presentation of results is of high quality. All steps are transparent and open, materials are provided online. I stated some of my concerns regarding reliability and paper selection in a previous section. The manuscript should be appropriately revised. Overall, the

statistical methods used, as well as data description and presentation are appropriate. The review is executed in a systematic and transparent manner.

Conclusions and data interpretation are robust, although the authors report themselves that the number of results available on causal inferences is small and should be interpreted with caution. This, of course, is not the authors' fault, as they can only review and analyze the data that is available.

Preregistration: Authors preregistered a protocol with the key research question and the search strategy applied, <https://osf.io/7ry4a/>

No deviations from the preregistration were reported.

References: The manuscript references previous literature appropriately and the literature review is well-organized and exhaustive. On page 6, end of second paragraph I suggest to cite literature on SoS so readers interested in this phenomenon know where to look (e.g., see the work by Scheufele & Moy <https://academic.oup.com/ijpor/article/12/1/3/739823?login=true> , the recent meta-analysis by Matthes et al., <https://journals.sagepub.com/doi/10.1177/0093650217745429>

Overall, this is a high-quality and relevant manuscript that just needs some more work. All the best with this research!

Reviewer #3:

Remarks to the Author:

I want to start this review by commending the authors for providing a well-written and thoughtful review of the literature on digital media and democracy, from which I gained substantial knowledge. I believe this is a valuable enterprise and I hope the criticism I will provide in this review will motivate the authors to strengthen their work.

My first concern with this review lies with the excessive simplification of the independent variable. The authors did as good a job as one can do in differentiating between the multiple outcomes that scholars of digital media and democracy have studied, but unfortunately, they failed to employ the same level of nuance and sophistication when they focused on the independent variable. Figure 4 provides a stark illustration of the problematic implications of this choice. Among the independent variables listed here one can find measures that range widely, too widely in my opinion, between generic and specific. Generic measures include social media use (twice), broadband availability (twice), internet use, and download speed. Specific measures include "I voted" messages (on Facebook), social media news use, partisan news exposure, offloading (?) political information, exposure to opposing views/counter-attitudinal news exposure. Somewhat in between are measures such as anti-refugee sentiment on Facebook and Twitter adoption by representatives. (And I have not even included the measures listed in the right hand-side of the figure which refer to studies of non-democratic regimes.) I look at all these different measures and I ask myself: What can we learn from this mumbo-jumbo of independent variables? How can we produce cumulative knowledge by comparing the effect of broadband availability – an aggregate-level outcome on which individuals have no control – with those of social media news use – an individual-level behavior that more often than not results from personal choices? As I read through the manuscript, I could not help but think about Giovanni Sartori's metaphor of the dog-cat (APSR 1970)—an animal that looks partly like a dog and partly like a cat, and is therefore impossible to meaningfully talk about. I am afraid this meta-analysis stretches the concept of the independent variable way too much for the

analyses to be useful towards cumulative knowledge. I appreciate that a meta-analysis needs to paint with broad brushes, but the brush used here to conceptualize and operationalize the independent variable is so broad that it ends up obscuring more than it reveals. If the authors could do the same fine job of differentiating the IV as they did for the DV, they could still present some top-line results similar to those shown in this manuscript, but then they could more precisely zoom in to more specific and meaningful conceptualizations and operationalizations of their key explanatory factors.

The second concern I have with this meta-analysis involves the file drawer problem. The authors dedicate a few lines to this issue in the "Limitations" section but I believe they have been slightly too cavalier about the problem. Why would studies of digital media and democracy suffer from this well-known problem less than others? I also worry that the authors' selection criteria, which are clearly illustrated, have contributed to this problem. One of Shelley Boulianne's meta-analyses that the study cites focused on digital media and political participation (one of the ten outcomes this manuscript addresses) and analyzed results from more than 300 studies. Yet, this review, which has a much broader focus, is limited to less than 500 studies. It is quite plausible that the focus on "quality" that transpires from the discussion of methods might have led the authors to overlook important but less editorially successful studies that reported null findings. Hence, I would encourage the authors to broaden the scope of the analysis---which might also help them increase the power of their analyses and, thus, be able to introduce the more nuanced distinctions of the IV I discussed earlier. I also think differentiating between pre-registered and non-pre-registered studies might help the authors estimate the extent to which the relative lack of null findings is indeed caused by the file drawer problem or not.

Finally, three small points.

First, I loved the figures included in the manuscript and want to commend the authors for doing such a good job with them. However, I found the color coding in Figure 1 to be confusing. If the colors are meant to represent increases or decreases in frequency, wouldn't it be easier to just use gradations on the same color, where darker tones represent higher frequencies and lighter tones lower frequencies?

Secondly, I was puzzled by Figure 3. It is a compelling representation of different strategies to make causal inferences, but it strongly felt like a detour from the main argument and analysis of the manuscript. I would encourage the authors to remove it, making space for more specific analyses of their own data.

Thirdly, while I appreciated the granularity with which the authors measured their DVs and the carefulness with which they analyzed some of the findings, I still feel some interpretations could be more nuanced. For instance, Diana Mutz (2006) has clearly shown that exposure to political talk one agrees with leads to higher levels of political participation, while exposure to disagreement depresses it. Trust in institutions can be beneficial to democracy, but the highest levels of trust are reported in authoritarian regimes, and many authors have argued that a skeptical but constructive, "trust but verify" attitude among citizens is more desirable for democracy than blind faith. Even populism comes in different shapes and sizes: is a vote for Spanish Podemos equally as problematic for democracy as a vote for the German AfD? Let me be clear that I do not consider these as fatal problem and I am not suggesting that the authors change their analysis to incorporate these issues, but I do believe that they could show a bit more awareness of these nuances as they interpret and discuss their findings.

I hope the authors find these comments helpful and wish them all the best in further developing this interesting research.

Author Rebuttal to Initial comments**Reviewer 1**

1. The authors should report the MOOSE guidelines checklist including page numbers to show where each guideline has been addressed in the manuscript. Without the checklist, it is not clear that these guidelines have been addressed.

Thank you for reminding us of this. We uploaded the completed MOOSE checklist to the OSF repository and pointed to it in the method section of our manuscript (see p. 20).

2. The pre-registration stated that the authors will assess study quality and sources of bias. Was this assessed and could the authors evaluate the evidence in line with the quality of the studies, and sources of bias?

In response to this important remark to assess study quality and risk of bias more deeply, we conducted an additional round of coding, along different dimensions of sample quality (size, sampling method and study type, see updated data_effects.xlsx) and risk of bias (pre-registration, open data and conflict of interest). We now present and discuss the results of this in the newly added section "Sampling methods and risk of bias" and the associated new figure 5 (see page 16). This additional information provides new insights, for example, into how distinct the outcome measures are that are examined using different types of data and sampling methods. We are thankful for this constructive comment.

3. The pre-registration stated that the authors will conduct and report a network analysis among the co-authors as well as co-citations of included papers. I could not see this in the systematic review. If the authors decided not to include this, they should explain why.

Thank you for raising this point. Actually, the co-author network was previously reported in the supplementary material (Figure S4), but in light of this comment we revisited this analysis and enhanced the network representation, so that edges indicate the direction of the reported association and the two graphs are split up between detrimental and beneficial outcomes. We could not create a meaningful co-citation network, due to the ambiguity of the exported data, which did not allow a good match of references. We noted that in the newly added section "deviations from the protocol" in the supplementary information (p. 1).

4. Study selection criteria: how were small-n lab experiments and small-N student surveys defined? I.e., what was the minimum N?

Of course the question of what sample size is considered "small" highly depends on the context. We decided to not consider relationships or effects of studies with $N < 100$. We made this more explicit now in the revised study selection criteria on p. 20 and the revised version of Figure 6.

5. Study selection criteria: The inclusion criteria statement (“To be included, articles had to include at least two distinct variables, treatment or independent variable and outcome”) wasn’t particularly specific. I recommend the authors omit the terms “treatment variable”, “independent variable” and “outcome” and replace them with the actual variables of interest.

Thank you for raising this point. We omitted the mentioned terms in the description of study selection criteria and described the procedure much more concretely (see p. 20)

6. Discussion: it seems unrealistic to suggest that publication bias is unlikely to exist because null effects of digital media on political outcomes would be important enough to report. While I agree that the null effects would be important, this is also the case for many other research questions where publication bias has been evidenced to exist. To examine potential publication bias / the file drawer problem, could the authors identify any pre-registered articles included in this review to see if they were more likely to report null results? And/or perhaps consult other quantitative meta-analyses on this or similar topics to see if they have identified publication bias?

We thank the reviewer for raising this important point and for their constructive suggestions. In response we did perform an additional round of coding to be able to estimate study quality (sampling methods, sample size and timing) and risk of bias (conflict of interest, pre-registration and open data) on all papers reporting one of the top outcome measures. The results of this analysis are now communicated in the additional section “Sampling methods and risk of bias” (see p. 15ff), in combination with the newly added Figure 5. We did not find clear indications of systematic biases in the results, because the patterns of reported associations stayed stable when varying a factor for the risk of bias e.g., between papers with and without open data or pre-registrations (unfortunately, the latter only comprised a very small set of papers). If at all, we found an asymmetry between studies that do not follow open data practices to report relatively more beneficial associations than those that do provide open data access. We also found that sampling methods are not uniformly distributed across outcome measures, with detrimental outcome measures more often being studies using behavioural data from social media, and that large, probabilistic samples were slightly more likely to report detrimental associations. On the author level, we revised the co-author network in Figure S4 to account for the reported associations and did not find a clear pattern beyond the expected author-communities. In addition, we contacted various researchers in the field (through prominent mailing lists, including one with nearly 700 members) about any unpublished results in this area, but received no unpublished papers in response. We believe that we were able to illuminate to the best of our possibilities the study quality, the risk of bias as well as the file-drawer problem, which will always remain, at least partly, inaccessible.

7. The authors should be commended for providing their dataset and figure code on the OSF. Two follow-up points on this: (1) Could the authors please state the name of the file that contains the dataset? E.g., is it "Data_effects_complete" or "data_final"? (2) It is stated that “code for the analyses and visualisations” is provided on the OSF, but only code used to derive the Figures is on the OSF. Was there any other code used to analyse data for the article? If so, could that be provided?

We agree that the file names in the OSF repository were lacking clarity. We now renamed them accordingly and describe the content more clearly in the data availability statement in the revised manuscript. Besides the full dataset

("full_data.xlsx") containing all collected 3534 papers, we provide two datasets: one, including the entire review sample (N = 499, "data_review.xlsx") and one including only those papers that examine at least one of the top 10 political outcome variables. In the latter dataset, not the publication but the effect reported is counted as a unit of observation. For example, if a paper reports two relationships (e.g. positive effect of digital media use on political knowledge but negative on trust), we find each relationship in a separate row of the dataset (N = 355, "data_effects.xlsx"). We stated the names of the datasets in our updated data availability statement including a link to the repository on page 23. Because the coding process in this review was purely a manual effort (besides simple, merging steps of tables), no code was used to generate those data files. However, we provide the updated code for all figures and subgroup analyses in the OSF repository.

8. The pre-registered question "If, to what degree and in which contexts, do digital media have detrimental effects on democracy?" seemed to be phrased in a complex way and include multiple components. I am not sure if it would be appropriate to edit it, given it was pre-registered, but if there is scope to do so, the authors could consider breaking it down into multiple questions, e.g: (1) "To what extent does digital media have detrimental effects on democracy?", and (2) "In what contexts does digital media have detrimental effects on democracy?".

Thank you very much for the suggestion. We agree that our initial phrasing of the research question is very complex and regret that we have not considered breaking it down at the stage of the preregistration. As the Editor suggested, we stick to the original phrasing as pre-registered, but we added a differentiation in the revised manuscript, as the Reviewer suggested. We now discuss both sub questions throughout the manuscript and furthermore motivate the context dependence in the revised introduction (see, e.g. p. 3, 6 and 13).

9. Data synthesis and analysis: The sentence "The final table of selected articles with coded variables will be published alongside this article as a major result of this review project" should be updated to provide the link to the dataset of selected articles with variables.

Thank you for reminding us of this point, we now state the names of the datasets in our updated data availability statement including a link to the repository on page 23.

10. Figure 2. For the insets, it is difficult to understand the labels – e.g., do they span multiple lines or is it one label per line? Also on the left hand side inset, there seem to be some labels with no data for either top or bottom panel – do they need to be included? Could these insets be stretched to show the labels more clearly? Or put as a separate figure to avoid detracting attention from the middle part, which is much easier to understand?

We thank the reviewer for raising these issues with Figure 2. In response, and regarding other remarks, we revised Figure 2 and show each inset with its own set of labels, so as to avoid any confusion. Also, those insets now show a different, more interesting breakdown along the different types of digital media, which we believe provides new insights beyond the previous version of the figure.

11. Figure 3. Three points: (1) Field experiments seem to be a common causal inference strategy in the included studies, but field experiments were not included in Fig 3, and it is unclear what they involved (from Fig. 4). I suggest adding a description of Field experiments to Fig 3. (2) There are many causal inference strategies other than

regression discontinuity that are not shown in the figure (e.g., twin studies, Mendelian randomisation, natural experiments), so perhaps rephrase the note to acknowledge this or omit it? (3) There is a small typo (“treatment”) in the “Matching” section.

Thank you for raising these important points. Let us reply to each point individually:

- (1) In general, we tried to keep our causal inference box streamlined, while also providing interested readers with an overview of causal inference strategies that are not highly familiar in many fields. We assumed that most readers are familiar with the experimental logic of identifying causal effects but might be less familiar with avenues to uncover causal effects in observational data. Indeed, field experiments conducted on social media platforms play a major role in our body of causal inference papers. While experiments, matching, instrumental variables and panel designs all serve to identify causal effects, we kept it outside the info-box because we saw one important distinction between (field) experiments and the other techniques mentioned. While experiments and field experiments target the data generation process (e.g. by actively inducing a treatment), the other strategies operate in the analysis or effect estimation step of the research process, often in the context of observational data. Therefore, we introduced the fundamental principle of causal inference (top of the info box) with the experimental logic of randomized treatment allocation. Even though the field experiments in our reviewed sample differ considerably, we saw that some of them also used instrumental-variable logic when recovering the intent to treat (ITT) effect in the data. We therefore mentioned this in our IV section of the info box: “IVs can be used in observational settings or in (field)experiments with imperfect compliance.”
- (2) This point is absolutely correct. We omit the note. Instead, we mentioned field experiments as the fourth major strategy in the note.
- (3) Thank you. We corrected the typo.

12. Figure 4. Some outcome terms were not clear, e.g. “correct voting” (what qualifies as correct?), “voting congruency” (congruent with what?), “turnout” (does this mean voter turnout?). I recommend the authors provide a table or further information defining the outcome terms.

Thank you for flagging this. We carefully reconsidered all outcome variables in Figure 3 (previously Figure 4) and refined the wording when unclear. For example, we replaced “turnout” with “voter turnout”, “voting congruency” with “attitudes of constituents reflected in representatives’ votes” and “correct voting” with “voting in line with personal attitudes”. In the methods section we do provide a frequency table of specific outcome measures in each category in Fig. 6d, but not specifically for the causal papers.

13. Figure 5. Fig 1b implies that 3-10 articles were from the UK, but the UK is not shown in Fig 5 – why was this?

Thank you for spotting this error, this was indeed an issue in the plotting code, where the list of countries on the y-axis is specified manually and we missed the UK. We corrected the code and updated Figure 4 (previously Figure 5) accordingly, thank you for finding this.

14. Fig. 6: Could the authors make it clearer which articles were referenced manually and were not identified in the original search?

With our updated flow-chart, we tried to break down the origins of the articles as transparently as possible. The first (green) number refers to the original search query that we ran in April 2021. The second (blue) number refers to the additional papers that we collected with exactly the same search query but that were published between April 2021 and September 2021. Only the (red) 22 papers were added manually to the review sample. In the course of doing the revisions, we came across one important causal paper (Geraci et al., 2022), that we decided to include manually in this stage. Therefore, the final sample increases from 498 to 499. All original 3534 papers with information on the origin (first or second query, or manual inclusion) as well as the decision stages (included, excluded based on the title, or excluded based on the abstract) can be found in the dataset “full_data.xlsx” on OSF.

Reviewer 2

Key results: Previous research in the field has examined effects on specific political variables like, for instance, political polarization, political participation, various forms of trust and political knowledge. This is the first review of its kind (to the best of my knowledge) to systematically review various political outcome variables together in an integrated manner. This manuscript substantially broadens our understanding regarding the effects of digital media. Examining political outcome variables in an integrated way can be regarded beneficial, as effects of digital media are complex affecting very different processes (at the same time), which may be deemed (from a normative perspective) both positive (i.e., political participation) and negative (i.e., political polarization). Looking at different outcome variables separately can certainly be informative. Yet, this review provides a broader perspective by examining very different outcome variables in an integrated way.

Also, it can help to disentangle different processes and clarify previous findings in the field, as the authors systematically differentiate between correlational and causal effects of digital media.

The results show both beneficial (e.g., effects on political participation) and detrimental effects (e.g., political polarization) for democracy and thus “ground for concern” (p. 16), as there is “clear evidence of serious threats to democracy” (p. 16), as the authors put it.

We thank the reviewer for the general, positive assessment of our work and its scope. We also hope that our work does exactly that, broadening our understanding of the effects of digital media. In the substantially revised version of our manuscript we hope to have underscored our results with additional rounds of coding that allows us to dive deeper into the digital media variables and the quality of studies, so to strengthen the methodological foundation of our finding, we respond to the individuals remarks below.

Validity: The manuscript follows a well-planned, transparent (i.e., preregistered) and well-executed procedure. First, one concern I have relates to the authors’ understanding of “digital media”, which is not explicitly defined and

operationalized. This should be changed. A clear definition should be provided early on in the manuscript, as the concept of “digital media” is obviously of key importance for the paper. Importantly, differences between “digital media” (in the authors’ understanding) and other types of media should be explained. On page 2 there is a hint. Authors may regard platforms that “do not create content” themselves as “digital media”. But this needs more explanation and justification. Also, on page 4 it is highlighted that authors do “not consider the effects of traditional media (e.g., television or radio)”. However, today such “traditional media contents” are regularly available online or on social media (i.e., “digital media”), be it on demand on platforms of traditional media outlets or online newspaper content that is spread via social media. In fact, a large proportion of “news” and links on social media stems from “traditional media outlets”, which move their content online (TV news available on YouTube; footage, links etc. can be shared on Twitter), radio programs that originally aired on national radio are then also available as a podcast on a commercial platform etc. All this being said, I would like to know how exactly the authors define “digital media” and how their understanding may have influenced the search process for articles and, in turn, the results. Would a different conceptualization of digital media potentially lead to different results?

I would like to emphasize that I do NOT regard the missing of a clear definition as a fundamental flaw or major problem that cannot be fixed. Instead, I would like to encourage authors to think about this aspect more, provide answers to my questions, and to revise the manuscript accordingly.

Thank you very much for raising this important point. You are absolutely correct that there are important differences between different digital media variables and that the simplification of digital media stands in contrast to our consideration of e.g. political outcome variables.

In order to fully account for this, we conducted a completely new round of coding in the full body of papers (N = 499) for a breakdown of the digital media variables and revised Figures 1, 2 and 3, as well as the corresponding text, in light of this newly obtained data. We believe that this dimension really opened up a few more insights. For example, that measures of selective exposure and homophilic social networks not only describe different outcome variables, but are also conducted in different areas of digital media, namely political information for the former and social media (especially Twitter) for the latter.

In more detail, for this additional coding round we followed the same procedure that we took for the political outcome variables. We visited the abstract of every single paper and noted the digital media variable while closely staying with the phrasing of the author of the respective paper. If not clearly mentioned in the abstract, we visited the full text of the manuscript. After that, we iteratively grouped those variables into categories and sorted single variables (e.g. Facebook news) to the best fitting larger group (e.g. political information on social media) until we reached a manageable overview of digital media variables (see new Figure 1d for the distribution of combinations between political outcomes and digital media variables). We also revised Figure 2 in light of those newly added variables, to show a break-down of the reported associations across those types of digital media. We saw that this uncovers a more informative insight than the previous version, namely that trust and polarization were measured in a much broader set of digital media, while news exposure and homophily in different and more narrow settings.

We also coded whether the digital media variable refers to social media, the internet more broadly, or whether the

digital media variable was so specific that it did not fit any of the two categories (“other”). We have split the results that are reported in Figure 2 along this dimension in the newly added supplementary figure S5, as well as split the result from causal papers in the revised version of Figure 3 (previously Figure 4).

Please find our description of the digital media categories on p. 3 of the manuscript, an overview of digital media variables in our sample in Figure 1d, and a differential consideration of our findings for different digital media categories on pages 9ff. and figures 2 and S5.

Second, I have some questions and concerns regarding the article selection process and its reliability. The search-query presented in Appendix A.1. seems to be well-developed and appropriate. My first question relates to the article exclusion procedure. Can the authors describe this process on page 18 a bit more? The number of articles was narrowed down from 3,518 to 1,351 to 741 and finally to 498. Authors provide some information on this procedure in the Materials and Methods section, but to me it remains unclear if coders/raters mutually agreed on study exclusion in these different steps. Did the authors measure this in any way? Do interrater-coefficients show acceptable results in regard to these first study exclusion steps?

Thank you for raising this point. Let us briefly explain the procedure. The first coding round, which ascertained whether a paper fits the review frame based on the title alone, was split between two coders. Coders could flag papers that are subject to discussion to let the other coder double check the decision. In this round, only papers that clearly did not fit were excluded from the sample. Criteria at this stage were:

- inclusion: original empirical studies of digital media in combination with a political outcome variable
- exclusion: theoretical papers, simulations, reviews, papers that clearly did not match the subject focus (e.g. papers on polarisation phenomena in physics), not English language (criteria that can be assessed by considering the title of a publication)

We would like to note that the query retrieved 3,512 papers and, in the initial version of the manuscript, we counted 6 manually added papers (that were added early in the process) to this initial sample. In the updated version of our flow chart in the SI, these 6 papers appear together with the 16 other manually (later in the process) added papers in the total of 22 manually added papers.

The next coding round considered abstracts and was again conducted in parallel by two coders. The inter-coder reliability, after this round of paper selection, was Krippendorff's alpha of 0.66 (87% agreement). After calculating this value, disagreement between coders was solved through discussion. At this stage, we excluded all studies that were not original empirical work such as other reviews or conceptual papers, simulation studies and purely methodological papers e.g. hate speech or misinformation detection approaches.

Afterwards, we conducted another, more in-depth coding round in which we refined our exclusion decisions (e.g. excluded studies that examined the digitisation of government, preprints, small-scale lab experiments, small-scale convenience or student samples and studies that only included one variable (e.g. description of online forums). In this round, also the parameters: variables, causality, method, political outcome, and moderation were extracted, using a

standardized extraction form. In a second coding round, also the digital media variable and category as well as potential sources of bias were coded. For an explanation of inter coder reliabilities at this stage, please consider our answer to your next point.

With our updated flow-chart, we tried to break down the origins of the articles as transparently as possible. The first (green) number refers to the original search query that we ran in April 2021. The second (blue) number refers to the additional papers that we collected with exactly the same search query but that were published between April 2021 and September 2021. Only the (red) 22 papers were added manually to the review sample. In the course of doing the revisions, we came across one important causal paper (Geraci et al., 2022), that we decided to include manually in this stage. Therefore, the final sample increases from 498 to 499. All original 3534 papers with information on the origin (first or second query, or manual inclusion) as well as the decision stages (included, excluded based on the title, or excluded based on the abstract) can be found in the dataset “full_data.xlsx” on OSF.

Third, one concern I have relates to the rather low Krippendorff’s alpha value of 0.66 reported on page 18. In the literature values in this range are rather critically discussed. Of course, it always depends on the particular research context/problem at hand meaning that coefficients have to be “put into study context”. Anyways, I would like to see a more critical discussion on this aspect of intercoder reliability and how it relates to your findings. What are potential limitations here? Also, I am not quite sure how your “overall” score of 87% relates to the Krippendorff and Cohen coefficients reported. Maybe, you can briefly explain this aspect. Finally, I would like to see a more fine-grained explanation of reliability coefficients in regard to the major categories that were coded, as reliability may be better in some categories compared to others and the overall reliability scores are thus not perfectly informative. Maybe authors can provide some more information and, for instance, insert a table (overview of reliability coefficients in regard to different categories).

This is an absolutely valid point and we regret that we did not discuss this more clearly. The value of Krippendorff’s Alpha presented in the text refers to the initial agreement on the selection of papers on the basis of the abstract between the two coders. Despite being low in other contexts, we do not consider the value to be particularly problematic, as we solved disagreement on the inclusion of a paper through discussion between the two coders.

The difference between percentage agreement and Krippendorff’s alpha is that percentage agreement looks at the plain ratios of observed agreements and disagreements (e.g. a scenario in which 100 papers were coded, both coders agreed on the codes for 90 but disagreed on 10, would result in a percentage agreement of 0.9 or 90%). By contrast, Krippendorff’s alpha is a chance-adjusted index that measures the amount of observed agreement and then adjusts it using an estimate of how much agreement would be expected by chance alone (e.g., through guessing).

We agree that a provision of inter-rater reliability coefficients for our key variables of interest would be helpful. However, it is important to note that when conducting a systematic review on such a huge body of evidence, categories cannot be defined prior to coding. Therefore, variable categories, especially for the outcome variables and digital media variables, were chosen inductively. In the first extraction step, coders stuck closely to the phrasing of the authors of the respective study. To reduce redundancy and refine the clustering of political outcome variables, digital media variables and methods, we iteratively generated frequency tables and manually sorted single variables to the best fitting categories until a small number of clearly distinct categories was selected. After the categories were

selected, both coders re-coded 10% of the sample to calculate inter coder reliabilities for all key variables. We now provide a table of inter coder reliabilities (percentage agreements and Krippendorff's Alphas) on p. 3 in the revised version of the supplementary information.

Originality and significance:

Based on an extensive review of research some of the key claims of the present manuscript are that there are both positive and negative effects of digital media on different political outcome variables. Six key areas of influence are identified: participation, trust, political knowledge, polarization, populism, network structures and news exposure.

Also, the authors see "clear evidence of serious threats to democracy". The findings suggest that much more research needs to study the role of "digital media" in other (non-Western) political contexts (e.g., authoritarian vs. democratic) and authors see a need for more research that examines causal effects of digital media on political outcome variables, as a lot of research in the field follows correlational approaches.

This paper significantly contributes to our understanding of digital media and effects on political behavior. The results are relevant and of interest to researchers and students in different disciplines (communication science, political science, psychology and intersecting fields such as political psychology) and many others interested in the question of how digital media affects politics and political behavior.

Again, we would like to thank the reviewer for the overall positive and constructive assessment of our work and hope that we could respond to their concerns satisfactorily below.

This being said, I suggest that authors explain the paper's purpose and innovation in an even better way. That is, "Why did the authors decide for this very broad methodological approach"? I can see the point and I generally welcome including multiple outcome variables to get an overall picture. Yet, at the same time this should be better explained and "justified", as there are also some downsides to this kind of approach (that should be explicitly named), which can be criticized. Why is one "overall review" better/needed (closes an important research gap?) compared to more focused reviews that focus on just one key dependent variable? I think, this needs to be much better justified and explained early on. Maybe authors can think about slightly re-structuring the sections on page 3/4 and say something about existing reviews and meta-analyses, why the present paper is important (why we need it), how it closes a key research gap and if other papers in the field followed a similar approach (or if the present paper is the first one in doing so).

We agree with the reviewer that the specific advantages of our approach as well as the distinction to existing reviews were not motivated well enough. As suggested, we did rewrite parts of the "Approach and Scope" section. In our additional analysis of different digital media types that we undertook in the course of this revision, another advantage of the comparison of different outcome variables became apparent, namely the different domains of digital media in which different outcomes are researched in. We believe that our revised manuscript thereby delivers another set of insights by showing further gaps in the literature of combinations of digital media and political variables. We hope that overall we could address this remark in our revision.

Overall, the paper presents an innovative and novel methodological approach and reports results that are important

and relevant to human behavior.

Data & methodology:

Authors present a valid and transparent approach of data selection and analysis. Presentation of results is of high quality. All steps are transparent and open, materials are provided online. I stated some of my concerns regarding reliability and paper selection in a previous section. The manuscript should be appropriately revised. Overall, the statistical methods used, as well as data description and presentation are appropriate. The review is executed in a systematic and transparent manner.

Conclusions and data interpretation are robust, although the authors report themselves that the number of results available on causal inferences is small and should be interpreted with caution. This, of course, is not the authors' fault, as they can only review and analyze the data that is available.

Preregistration: Authors preregistered a protocol with the key research question and the search strategy applied, <https://osf.io/7ry4a/>
No deviations from the preregistration were reported.

Again, we thank the reviewer for their assessment and believe to have added further transparency in the exclusion process by including the full set of papers in the OSF as well as a more detailed description of the process in the Methods section.

References: The manuscript references previous literature appropriately and the literature review is well-organized and exhaustive. On page 6, end of second paragraph I suggest to cite literature on SoS so readers interested in this phenomenon know where to look (e.g., see the work by Scheufele & Moy <https://academic.oup.com/ijpor/article/12/1/3/739823?login=true> , the recent meta-analysis by Matthes et al., <https://journals.sagepub.com/doi/10.1177/0093650217745429>

As suggested, we added the pointers to the literature on SoS, see page 7 of the revised manuscript.

Overall, this is a high-quality and relevant manuscript that just needs some more work. All the best with this research!

Thank you again for the constructive and helpful remarks that helped to improve the paper.

Reviewer 3

My first concern with this review lies with the excessive simplification of the independent variable. The authors did as good a job as one can do in differentiating between the multiple outcomes that scholars of digital media and democracy have studied, but unfortunately, they failed to employ the same level of nuance and sophistication when they focused on the independent variable. Figure 4 provides a stark illustration of the problematic implications of this

choice. Among the independent variables listed here one can find measures that range widely, too widely in my opinion, between generic and specific. Generic measures include social media use (twice), broadband availability (twice), internet use, and download speed. Specific measures include “I voted” messages (on Facebook), social media news use, partisan news exposure, offloading (?) political information, exposure to opposing views/counter-attitudinal news exposure. Somewhat in between are measures such as anti-refugee sentiment on Facebook and Twitter adoption by representatives. (And I have not even included the measures listed in the right hand-side of the figure which refer to studies of non-democratic regimes.) I look at all these different measures and I ask myself: What can we learn from this mumbo-jumbo of independent variables? How can we produce cumulative knowledge by comparing the effect of broadband availability – an aggregate-level outcome on which individuals have no control – with those of social media news use – an individual-level behavior that more often than not results from personal choices? As I read through the manuscript, I could not help but think about Giovanni Sartori’s metaphor of the dog-cat (APSR 1970)—an animal that looks partly like a dog and partly like a cat, and is therefore impossible to meaningfully talk about. I am afraid this meta-analysis stretches the concept of the independent variable way too much for the analyses to be useful towards cumulative knowledge. I appreciate that a meta-analysis needs to paint with broad brushes, but the brush used here to conceptualize and operationalize the independent variable is so broad that it ends up obscuring more than it reveals. If the authors could do the same fine job of differentiating the IV as they did for the DV, they could still present some top-line results similar to those shown in this manuscript, but then they could more precisely zoom in to more specific and meaningful conceptualizations and operationalizations of their key explanatory factors.

Thank you very much for raising this important point. You are absolutely correct that there are important differences between different digital media variables and that the simplification of digital media stands in contrast to our consideration of e.g. political outcome variables.

In order to fully account for this, we conducted a completely new round of coding in the full body of papers (N = 499) for a breakdown of the digital media variables and revised Figures 1, 2 and 3, as well as the corresponding text, in light of this newly obtained data. We believe that this dimension really opened up a few more insights. For example, that measures of selective exposure and homophilic social networks not only describe different outcome variables, but are also conducted in different areas of digital media, namely political information for the former and social media (especially Twitter) for the latter.

In more detail, for this additional coding round we followed the same procedure that we took for the political outcome variables. We visited the abstract of every single paper and noted the digital media variable while closely staying with the phrasing of the author of the respective paper. If not clearly mentioned in the abstract, we visited the full text of the manuscript. After that, we iteratively grouped those variables into categories and sorted single variables (e.g. Facebook news) to the best fitting larger group (e.g. political information on social media) until we reached a manageable overview of digital media variables (see new Figure 1d for the distribution of combinations between political outcomes and digital media variables). We also revised Figure 2 in light of those newly added variables, to show a break-down of the reported associations across those types of digital media. We saw that this uncovers a more informative insight than the previous version, namely that trust and polarization were measured in a much broader set of digital media, while news exposure and homophily in different and more narrow settings.

We also coded whether the digital media variable refers to social media, the internet more broadly, or whether the digital media variable was so specific that it did not fit any of the two categories (“other”). We have split the results that are reported in Figure 2 along this dimension in the newly added supplementary figure S5, as well as split the result from causal papers in the revised version of Figure 3 (previously Figure 4).

Please find our description of the digital media categories on p. 3 of the manuscript, an overview of digital media variables in our sample in Figure 1d, and a differential consideration of our findings for different digital media categories on pages 9ff and figures 2 and S5.

The second concern I have with this meta-analysis involves the file drawer problem. The authors dedicate a few lines to this issue in the “Limitations” section but I believe they have been slightly too cavalier about the problem. Why would studies of digital media and democracy suffer from this well-known problem less than others? I also worry that the authors’ selection criteria, which are clearly illustrated, have contributed to this problem. One of Shelley Boulianne’s meta-analyses that the study cites focused on digital media and political participation (one of the ten outcomes this manuscript addresses) and analyzed results from more than 300 studies. Yet, this review, which has a much broader focus, is limited to less than 500 studies. It is quite plausible that the focus on “quality” that transpires from the discussion of methods might have led the authors to overlook important but less editorially successful studies that reported null findings. Hence, I would encourage the authors to broaden the scope of the analysis---which might also help them increase the power of their analyses and, thus, be able to introduce the more nuanced distinctions of the IV I discussed earlier. I also think differentiating between pre-registered and non-pre-registered studies might help the authors estimate the extent to which the relative lack of null findings is indeed caused by the file drawer problem or not.

We thank the reviewer for raising this important point and for their constructive suggestions. While we did not broaden the inclusion criteria, as we did not want to deviate from the pre-register process, we did perform an additional round of coding to be able to further estimate study quality (sampling methods, sample size and timing) and risk of bias (conflict of interest, pre-registration and open data) on all papers reporting one of the top outcome measures. The results of this analysis are now communicated in the additional section “Sampling methods and risk of bias” (see p. 15), in combination with the newly added Figure 5. We did not find clear indications of systematic biases in the results, because the patterns of reported associations stayed stable when varying a factor for the risk of bias e.g., between papers with and without open data or pre-registrations (unfortunately, the latter only comprised a very small set of papers). If at all, we found an asymmetry between studies that do not follow open data practices to report relatively more beneficial associations than those that do provide open data access. We also found that sampling methods are not uniformly distributed across outcome measures, with detrimental outcome measures more often being studies using behavioural data from social media, and that large, probabilistic samples were slightly more likely to report detrimental associations. On the author level, we revised the co-author network in Figure S4 to account for the reported associations and did not find a clear pattern beyond the expected author-communities. In addition, we contacted various researchers in the field (through prominent mailing lists, including one with nearly 700 members) about any unpublished results in this area, but received no unpublished papers in response. We believe that we were able to illuminate to the best of our possibilities the study quality, the risk of bias as well as the file-drawer problem,

which will always remain, at least partly, inaccessible. We hope that we could address the concerns articulated here with this set of steps in our revision.

Finally, three small points.

First, I loved the figures included in the manuscript and want to commend the authors for doing such a good job with them. However, I found the color coding in Figure 1 to be confusing. If the colors are meant to represent increases or decreases in frequency, wouldn't it be easier to just use gradations on the same color, where darker tones represent higher frequencies and lighter tones lower frequencies?

In Figure 1 we used the color palettes that are recommended for sequential data to offer the best compromise between being color-blind friendly, still visible in black and white print out (<https://matplotlib.org/3.1.0/tutorials/colors/colormaps.html>). But if needed we can still switch the color palettes to simpler gradients.

Secondly, I was puzzled by Figure 3. It is a compelling representation of different strategies to make causal inferences, but it strongly felt like a detour from the main argument and analysis of the manuscript. I would encourage the authors to remove it, making space for more specific analyses of their own data.

We agree with the reviewer that Figure 3 is not very specific about our data and just serves as an overview, accordingly we moved it into the methods section (now Figure 7).

Thirdly, while I appreciated the granularity with which the authors measured their DVs and the carefulness with which they analyzed some of the findings, I still feel some interpretations could be more nuanced. For instance, Diana Mutz (2006) has clearly shown that exposure to political talk one agrees with leads to higher levels of political participation, while exposure to disagreement depresses it. Trust in institutions can be beneficial to democracy, but the highest levels of trust are reported in authoritarian regimes, and many authors have argued that a skeptical but constructive, "trust but verify" attitude among citizens is more desirable for democracy than blind faith. Even populism comes in different shapes and sizes: is a vote for Spanish Podemos equally as problematic for democracy as a vote for the German AfD? Let me be clear that I do not consider these as fatal problem and I am not suggesting that the authors change their analysis to incorporate these issues, but I do believe that they could show a bit more awareness of these nuances as they interpret and discuss their findings.

We have expanded our justification of our broad approach better in the revised manuscript and revised the disclaimer about the lack of distinction within dimensions under research and hope that the reviewer's concerns are covered by that.

Let us conclude with a thank you to the reviewer for the positive and constructive feedback that helped to advance our manuscript further.

Decision Letter, first revision:

22nd June 2022

Dear Dr. Lorenz-Spreen,

Thank you for submitting your revised manuscript "Digital Media and Democracy: A Systematic Review of Causal and Correlational Evidence Worldwide" (NATHUMBEHAV-211217412A). It has now been seen by the original reviewers, and all reviewer feedback is included at the end of this letter. You will see from their comments that the reviewers are divided in their views of your revision. Reviewers 1 and 2 were satisfied with the changes made. In contrast, Reviewer 3 indicated that their previous concerns have not been addressed, and the sample of studies misses important papers and doesn't sufficiently cover all of the outcomes included.

In light of this split among the reviewers, we consulted with Reviewer 2 regarding Reviewer 3's comments and whether they shared the concerns raised. Reviewer 2 indicated that they do agree with Reviewer 3 that the study inclusion process must be transparently explained and limitations noted. However, they also indicated that they view the inclusion of papers across many domains as a strength of your study, and they did not feel that there was systematic exclusion of relevant work.

After careful consideration of all reviewer feedback, we will be happy in principle to publish your study in Nature Human Behaviour, pending revisions to satisfy the referees' final requests and to comply with our editorial and formatting guidelines.

We are now performing detailed checks on your paper and will send you a checklist detailing our editorial and formatting requirements within a week. Please do not upload the final materials and make any revisions until you receive this additional information from us.

Sincerely,
Aisha

Aisha Bradshaw, PhD
Senior Editor
Nature Human Behaviour

Reviewer #1 (Remarks to the Author):

The authors have done an excellent job in revising the manuscript and I have no further suggestions. This is a very important and comprehensive review that will make an excellent contribution to the literature.

Reviewer #2 (Remarks to the Author):

The authors did a very good job in revising the manuscript. All my points very addressed, my original concerns alleviated. Especially, I think the newly integrated information on papers selection and the coding process increase clarity and general understanding. Good luck with your research.

Reviewer #3 (Remarks to the Author):

It was a pleasure to read and evaluate the revised version of this manuscript. I want to commend the authors for thoughtfully engaging with the detailed reviews they received, including my own. I continue to believe there is merit in this study, but unfortunately, I also continue to believe that there are two major flaws in its execution that, in my opinion, do not make it suitable for publication in Nature Human Behavior.

In my review of the first version of the manuscript, I emphasized what I considered as two major problems. The first (which another reviewer also noted) was the lumping together of many different independent variables measuring different aspects of digital/social media at the aggregate and individual levels. The second was the small number of studies included in the analysis (499 in total across ten different political outcomes). In my original review I had noted that "One of Shelley Boulianne's meta-analyses that the study cites focused on digital media and political participation (one of the ten outcomes this manuscript addresses) and analyzed results from more than 300 studies."

The authors did as good a job as they could do, within the limitations of their sample of studies, to address the first problem. The additional analyses they conducted do shed some additional light on the specific types of digital/social media that have been studied as independent variables and how these relate to both the outcomes that have been studied and whether the outcomes were positive or negative for democracy. However, most of the analyses and discussion thereof are still focused on the original "lumpy" approach that combines all these independent variables into one. For instance, Figure 2 makes it clear that the key takeaways are those in the central part, which lumps together different measures, while offering a more refined breakdown of the independent variables for four out of ten of the political outcomes. Figure 3 only offers a crude distinction between "internet" and "social media". Figure 4 lumps together all measures. While I do appreciate the authors' efforts to shed further light on these aspects, I do not feel these improvements go far enough and I continue to worry that the way the key concepts have been stretched for analytical purposes greatly reduces the contribution of this study.

The authors acknowledged the second point in their response, but they chose to strictly follow their pre-registration inclusion criteria, which means that the empirical basis of the analysis remains too thin, in my view, to be credible. I am sorry to have to repeat myself, but a meta-analysis that focuses on ten different outcomes on the basis of less than twice the amount of studies covered by a previously published meta-analysis on just one of these outcomes will not pass muster with readers who are knowledgeable of the field and of the wide variety of studies that are constantly being published about the relationship between digital media and democratically relevant outcomes. I appreciate that the authors wish to follow the precepts of open science, but in my view this is a case when this laudable commitment stands in the way of actually doing the best possible science, and I

am afraid I find fault with the outcome, even if it resulted from good intentions and good practices, because it is simply not credible as a portrayal of the vast amount of research published on these topics. I wish the authors had piloted their study identification protocols and benchmarked the results against Boulianne's and other meta-analyses before pre-registering them, but unfortunately this does not seem to have been the case. I support open science, but I am not of the view that suboptimal choices should be placed outside the boundaries of criticism because they have been pre-registered, and unfortunately it is my belief that this has been the case with this study.

I also appreciate that the authors did what they could to address some of the limits of their dataset in their regard. The additional analyses they conducted where they looked at open data, pre-registration, and conflicts of interest declarations are informative, but as the authors acknowledge themselves, the numbers of preregistered studies and conflicts of interest declarations are too small to warrant any robust conclusions. The revised manuscript has retained the statement that the file drawer problem should be less of an issue for this kind of research than for other fields, which I continue to find unsubstantiated and unhelpful.

In sum, I commend the authors for their hard work and rigorous approach to an important set of problems, but I continue to believe that this revised manuscript, in spite of some improvements, suffers from the two major flaws I had identified in my first review, and for this reason my judgment remains that this manuscript is not suitable to be published in Nature Human Behavior. I wish the authors all the best in developing their important research.

Author Rebuttal, first revision

We thank the reviewers for their constructive remarks and their positive evaluations, the process helped to improve the manuscript considerably. Below we respond to the remaining points from Reviewer #3 in blue.

Reviewer #1 (Remarks to the Author):

The authors have done an excellent job in revising the manuscript and I have no further suggestions. This is a very important and comprehensive review that will make an excellent contribution to the literature.

Reviewer #2 (Remarks to the Author):

The authors did a very good job in revising the manuscript. All my points very addressed, my original concerns alleviated. Especially, I think the newly integrated information on papers selection and the coding process increase clarity and general understanding. Good luck with your research.

Reviewer #3 (Remarks to the Author):

It was a pleasure to read and evaluate the revised version of this manuscript. I want to commend the authors for thoughtfully engaging with the detailed reviews they received, including my own. I continue to believe there is merit in this study, but unfortunately, I also continue to believe that there are two major flaws in its execution that, in my opinion, do not make it suitable for publication in Nature Human Behavior.

In my review of the first version of the manuscript, I emphasized what I considered as two major problems. The first (which another reviewer also noted) was the lumping together of many different independent variables measuring different aspects of digital/social media at the aggregate and individual levels. The second was the small number of studies included in the analysis (499 in total across ten different political outcomes). In my original review I had noted that “One of Shelley Boulianne’s meta-analyses that the study cites focused on digital media and political participation (one of the ten outcomes this manuscript addresses) and analyzed results from more than 300 studies.”

We appreciate the reviewer’s follow-up comments on our consideration of digital media variables as well as our sample and selection criteria. Below we respond to both points in more detail and lay out that those research decisions were made deliberately and carefully to reach the goal of a rigorous and selective systematic review that summarizes a broad, emerging field across disciplines. More specifically, we chose to search broadly on both political and digital media variables, in order to capture the breadth of the literature and findings, which we explicitly discuss as a deliberate trade-off in our manuscript (see section “Limitations”).

The second concern, regarding the number of papers in our sample is a consequence of our requirements on the studies. We selected 499 from a pool of 3,518 articles. We applied and pre-registered a quite rigorous search strategy that did rely on Web of Science and Scopus, not primarily on Google Scholar (as recommended for systematic reviews here <https://onlinelibrary.wiley.com/doi/full/10.1002/jrsm.1378>), by that we include only published work and no pre-prints, for example. In addition, our strict selection criteria, which are listed in Supplementary Table 1, exclude laboratory experiments and other studies that do not directly address the specific research question of the status-quo of social media effects in an ecological setting.

The authors did as good a job as they could do, within the limitations of their sample of studies, to address the first problem. The additional analyses they conducted do shed some additional light on the specific types of digital/social media that have been studied as independent variables and how these

relate to both the outcomes that have been studied and whether the outcomes were positive or negative for democracy. However, most of the analyses and discussion thereof are still focused on the original “lumpy” approach that combines all these independent variables into one. For instance, Figure 2 makes it clear that the key takeaways are those in the central part, which lumps together different measures, while offering a more refined breakdown of the independent variables for four out of ten of the political outcomes. Figure 3 only offers a crude distinction between “internet” and “social media”. Figure 4 lumps together all measures. While I do appreciate the authors’ efforts to shed further light on these aspects, I do not feel these improvements go far enough and I continue to worry that the way the key concepts have been stretched for analytical purposes greatly reduces the contribution of this study.

In response to the reviewer’s observation, we would like to emphasize that all information, both, on the binary classification between social media and internet more broadly, and on the fine-grained categories of digital media variables reported by the authors of the studies can be found in the meta-dataset of studies (which can be found here <https://osf.io/9ab7n>) that we regard as a major outcome of our systematic review and as the core contribution to the research literature. We would also like to point to the necessity to reduce complexity in the dimensionality of some variables to permit transparent and comprehensible visualization.

Furthermore, as reviewer 3 noted correctly, we do present the exact breakdown of relationship directions for the fine grained digital media categories of four central political outcomes in Figures 1 and 2. Considering the patterns visible here, we do not observe any systematic differences (such as, for instance, mostly beneficial relationships for one digital media type but mostly detrimental relationships for another). The differential patterns we do find, namely more detrimental relationships for political knowledge and political expression in the context of social media when compared to the Internet more broadly (see Figure S5), are discussed.

In other words, a review or meta-analysis inevitably has to make a choice about the granularity of the presentation. At one extreme, 'lumping together' is avoided by presenting each study in isolation. At the other extreme, all differences between studies would be ignored and they would be 'lumped together' in a single data point. Neither extreme is usually appropriate. Instead, a systematic review must exercise judgment by aggregating across studies with only minor variations in methodology while simultaneously differentiating between clusters that are conceptually different. It is precisely for those reasons that Figures 2 and 3 differentiate between meaningfully different clusters. In our judgment, this is neither crude nor does it 'lump' studies together, but it represents the conceptually most appropriate lens through which to view the results. Readers who wish to delve into further differentiation can do so by inspecting the data table in the repository. The additional coding of digital media variables as a response to the reviewer’s remarks added important dimensionality, now reflected in the data table Figure 1 and Figure 2.

The authors acknowledged the second point in their response, but they chose to strictly follow their pre-registration inclusion criteria, which means that the empirical basis of the analysis remains too thin, in my view, to be credible. I am sorry to have to repeat myself, but a meta-analysis that focuses on ten different outcomes on the basis of less than twice the amount of studies covered by a previously published meta-analysis on just one of these outcomes will not pass muster with readers who are knowledgeable of the field and of the wide variety of studies that are constantly being published about the relationship between digital media and democratically relevant outcomes. I appreciate that the authors wish to follow the precepts of open science, but in my view this is a case when this laudable commitment stands in the way of actually doing the best possible science, and I am afraid I find fault with the outcome, even if it resulted from good intentions and good practices, because it is simply not credible as a portrayal of the vast amount of research published on these topics. I wish the authors had piloted their study identification protocols and benchmarked the results against Boulianne's and other meta-analyses before pre-registering them, but unfortunately this does not seem to have been the case. I support open science, but I am not of the view that suboptimal choices should be placed outside the boundaries of criticism because they have been pre-registered, and unfortunately it is my belief that this has been the case with this study.

Contrary to the reviewer's claim, our preregistration explicitly allowed us to include relevant studies that were not captured by our automated query. We made use of this option after we compared our dataset of included studies with the datasets of other relevant reviews in the field. One of these reviews was indeed Boulianne's review on social media and political participation. The factors leading to the seemingly small set of studies in our review is our inclusion process and its stricter inclusion criteria. For instance, we excluded preprints and we adopted a stricter definition of a minimal sample size. This means that we excluded studies that had (then) not yet passed the review process and that are likely to have produced results of lower reliability.

I also appreciate that the authors did what they could to address some of the limits of their dataset in their regard. The additional analyses they conducted where they looked at open data, pre-registration, and conflicts of interest declarations are informative, but as the authors acknowledge themselves, the numbers of preregistered studies and conflicts of interest declarations are too small to warrant any robust conclusions. The revised manuscript has retained the statement that the file drawer problem should be less of an issue for this kind of research than for other fields, which I continue to find unsubstantiated and unhelpful.

We thank the reviewer for this comment. Despite our best efforts to quantify indicators of the file drawer problem, we cannot make a concluding statement here. We therefore have decided now to drop the statement to which the reviewer objects.

In sum, I commend the authors for their hard work and rigorous approach to an important set of problems, but I continue to believe that this revised manuscript, in spite of some improvements, suffers from the two major flaws I had identified in my first review, and for this reason my judgment remains that this manuscript is not suitable to be published in Nature Human Behavior. I wish the authors all the best in developing their important research.

Final Decision Letter:

Dear Philipp,

We are pleased to inform you that your Article "A Systematic Review of Worldwide Causal and Correlational Evidence on Digital Media and Democracy", has now been accepted for publication in Nature Human Behaviour.

Please note that *Nature Human Behaviour* is a Transformative Journal (TJ). Authors whose manuscript was submitted on or after January 1st, 2021, may publish their research with us through the traditional subscription access route or make their paper immediately open access through payment of an article-processing charge (APC). Authors will not be required to make a final decision about access to their article until it has been accepted. IMPORTANT NOTE: Articles submitted before January 1st, 2021, are not eligible for Open Access publication. Find out more about Transformative Journals

With best regards,
Aisha

Aisha Bradshaw, PhD
Senior Editor
Nature Human Behaviour